# Doubling of surface oceanic meridional heat transport by non-symmetry of mesoscale eddies

Hailin Wang [1], Bo Qiu[2], Hanrui Liu [1] & Zhengguang Zhang [1,3] ✉

Oceanic transport of heat by ubiquitous mesoscale eddies plays a critical role in regulating climate variability and redistributing excess heat absorbed by ocean under global warming. Eddies have long been simplified as axisymmetric vortices and their influence on heat transport remains unclear. Here, we combine satellite and drifter data and show that oceanic mesoscale eddies are asymmetric and directionally-dependent, and are controlled by their self-sustaining nature and their dynamical environment. Both the direction and amplitude of eddy-induced heat fluxes are significantly influenced by eddy's asymmetry and directional dependence. When the eddy velocity field is decomposed into asymmetric and symmetric components, the eddy kinetic energy exhibits a nearly equal partition between these two components. The total eddy-induced meridional heat flux similarly doubles the heat flux induced by the symmetric components, highlighting the crucial contribution of eddy asymmetry on the magnitude of eddy-induced oceanic heat transport.

Global warming is one of the most critical challenges facing humanity today and the urgency to understand and predict the global climate change has become increasingly important. More than 90% of the heat accumulation in climate system of the past five decades has been in the ocean[1] and, as such, ocean plays a crucial role in determining the climate of the Earth and acts as a "buffer" against global warming due to its large heat capacity[2,3]. It is a long-standing and key issue to understand how the heat is transported and redistributed in the ocean. Observations show the temperature increase of the ocean surface is not homogenous but has great spatial diversity, such as enhanced warming in the western boundary current systems[4,5]. This diversity induced by heat redistribution has resulted in significant variations in regional responses to the global warming, causing extreme events such as drought, floods, storminess, wildfires and marine heat waves[6].

Mesoscale eddies with a horizontal scale of tens to hundreds of kilometers are ubiquitous in the ocean, accounting for nearly 90% of oceanic kinetic energy[7,8] and inducing significant transports of heat, salt, dissolved $CO_2$ and other tracers, comparable in magnitude with those of the large-scale wind- and thermohaline-driven circulation[9–13].

In some of the mid-latitude regions, eddy-induced heat transport accounts for nearly half of the total oceanic heat transport[14–17]. Thus, oceanic eddies can play a crucial role in setting the climatological mean state of the ocean, inducing variations with interannual to decadal time scales, and redistributing heat in the ocean under the global warming scenario[18–20].

Nowadays, prediction of future global warming and climate change relies mainly on numerical simulation by climate models. Simulations suggest that the presence or absence of mesoscale eddies in models could lead to a difference of 3 °C in 100-years global warming projection[21]. As such, it is impossible to obtain a reliable prediction of global climate change without proper simulation of oceanic eddies. A major effort to improve the simulation of climate model is to increase the model resolution and directly resolve eddies, which lead to improvements of the simulation of climate variability and the meridional heat fluxes[22–24]. However, the simulated eddy kinetic energy (EKE) can deviate from observations by about 50%-100%, leading to substantial errors of the eddy-induced heat fluxes in these eddy-resolving climate models[24–26]. On the other hand, most

[1]Frontiers Science Center for Deep Ocean Multispheres and Earth System (FDOMES) and Key Laboratory of Physical Oceanography, Academy of the Future Ocean, Chongben Honors College, Ocean University of China, Qingdao, PR China. [2]Department of Oceanography, University of Hawaii at Manoa, Honolulu, HI, USA. [3]Laoshan Laboratory, Qingdao, PR China. ✉e-mail: zhengguang@ouc.edu.cn

ocean components of the nowadays climate models, such as in the sixth phase of the Coupled Model Intercomparison Project (CMIP6), have to use a coarse resolution (~60 km) that cannot resolve mesoscale eddies and have to parameterize the eddy-induced transport[27–29]. According to the recent trend of computing capacity, this situation will not be substantially improved within the next five years to even a decade, and parameterization of eddy induced flux will still be indispensable in the near future[30,31]. Downgradient diffusion along the isopycnals of tracers and layer-thickness has been considered by the Gent-McWilliams scheme and other parameterizations[32,33]. Even the anisotropic diffusion has been justified both by laboratory and numerical experiments, which are influenced by planetary beta effect, the background flow field and topographic features[34–39]. In contrast, the global observational constraint useful for improvements of eddy parameterizations remains rather rare nowadays.

Despite the advent of satellite altimeters, surface drifters, Argo floats and other observational platforms in the past three decades that have led to significant advances in our understanding of the spatial structure and temporal evolution of mesoscale eddies[40–44], oceanic eddies have long been treated as axis-symmetric circular vortices for simplicity[10,12,19,41,45]. When eddies emerge from an onset of instability relating to the structure of potential vorticity (PV) gradient, their structures tend to have an asymmetric nature. Due to the $\beta$ effect, most of the mesoscale eddies move westward as "blobby" structures[40,41,46], which also break the symmetry of eddies. Observations of satellite altimetry and surface drifters confirm that the mesoscale eddies are commonly elongated along certain directions in the global ocean. This asymmetry exists ubiquitously and the observed eddies have been shown with statistical significance to differ from a perfect circle[47–49], although the controlling factors of the eddy asymmetry remains unknown. Mesoscale eddies can induce horizontal heat flux by trapping and stirring processes, which commonly correspond to the monopole and dipole structure of the surface temperature tracer field, respectively[16,50,51]. But how the asymmetry of eddy's dynamical structure (i.e., velocity and pressure field) influence the total eddy-induced heat transports remains unclear.

Here we show that there exists a close relationship between the asymmetry of mesoscale eddies and the eddy-induced heat transport. Based on observational data with a global coverage, we evaluate the dynamical factors controlling the asymmetry and directional-dependence of mesoscale eddies from a statistical point of view. The magnitude of eddy-induced heat transport is found to be significantly influenced by the asymmetry and directional-dependence of eddies. Furthermore, we demonstrate that the *asymmetric* part of the eddy flow field can induce a meridional heat flux almost as large as that by its symmetric counterpart. In other words, the actual eddy-induced heat flux is nearly doubled compared with its axisymmetry-induced contribution.

## Results

### Asymmetry and directional-dependence of mesoscale eddy

Consistent with the classical physical picture of a vortex, the solution of an isolated mesoscale eddy on a uniformly rotating Earth tends to be an axisymmetric vortex[52,53]. Meanwhile, vortices possess the self-sustaining ability and attain axisymmetry. This axisymmetrization process occurs naturally to restore the circular shape of a vortex when it is subjected to external perturbations under conditions of weak background gradients[54]. In quasi-geostrophic turbulence, the final stage often results in multiple sparsely-distributed eddies with approximately circular shapes[55,56]. Thus, mesoscale oceanic eddies have long been treated as circular in simple theories.

Energetic mesoscale eddies are ubiquitous in the ocean and serve as a principal sink for energy of planetary-scale mean oceanic circulation through "balanced" instabilities, e.g. the quasi-geostrophic barotropic and baroclinic instabilities[57]. Consequently, eddies are constantly interacting with large-scale motions. Due to their dense distribution, eddy-eddy interactions persist throughout the life cycle of mesoscale eddies. They are thus neither free-evolving nor in a sparsely-distributed, nearly isolated state in the actual ocean. Shear and strain induced by large-scale circulation and nearby eddies make it difficult for an effective axisymmetrization. Furthermore, potential vorticity that describes the rotational effect of stratified fluid is not homogenous due to the latitudinal dependence of the Coriolis parameter. The meridional gradient of the planetary PV establishes an anisotropic dynamical environment for mesoscale eddies, which fundamentally breaks the axisymmetry of eddies[58]. As a result, circular eddies are rather rare in the observations, and most of the eddies are non-axisymmetric, as shown by the representative sea surface height anomaly (SSHA) map in Fig. 1a.

In order to evaluate the level of eddies' non-axisymmetry, $I_a = a/b$ is introduced as a non-axisymmetry index, where $a$ is the length of major axis, and $b$ is the length of minor axis (Fig. 1b, c). As shown in Fig. 2a, the non-axisymmetry index $I_a$ has an average value of 1.55, which suggests that the major axes of the eddies are about 55% larger than their minor axes as a statistical average. Consequently, majority of the eddies exhibit an elongated shape along the direction of their major axes. As shown in Fig. 2b, the global distribution of $I_a$ indicates that asymmetry of eddies is more pronounced in three types of regions: Firstly, in the low latitude regions; secondly, in the regions with strong large-scale currents, such as Kuroshio, Gulf Stream and Antarctic Circumpolar Current (ACC); and thirdly, in the regions around major bottom topographic features or along continental boundaries. These regions share a common feature of strong gradient of the planetary PV, the large-scale stratification or the bottom topography, which suggests that larger background gradient can result in stronger asymmetry of eddies. From a dynamical point of view, background PV gradient tends to squeeze the quasi-geostrophic motions in a down/up gradient direction and elongates the motions along the background PV contours; e.g., Rossby waves can be zonally elongated by the planetary PV gradient and form zonal jets[58]. Therefore, it is natural to expect that eddies with larger horizontal scales are more strongly influenced by the background PV gradient and this is confirmed by observations that eddies with greater radii exhibit stronger non-axisymmetry and larger $I_a$ values (Supplementary Fig.S4). Considering the planetary PV gradient is not uniform globally, the Rhines scale $L_R = 2\pi\sqrt{2U/\beta}$ is introduced, where $U$ is the amplitude of eddy rotational speed and $\beta$ is the meridional gradient of Coriolis parameter $f$. When the horizontal scale of a quasi-geostrophic motion is larger than Rhines scale $L_R$, it will be significantly influenced by the background planetary PV gradient[58]. Thus, the normalized eddy radius by Rhines scale $R_n = R/L_R$ can serve as a dynamical index on whether an eddy is sufficiently large to feel $\beta$ effect. Consistent with our expectations, Fig. 2c reveals that eddies tend to be more non-axisymmetric with larger $R_n$.

Beside the background factors, self-sustaining eddy could preserve its shape in a weak background PV gradient. This ability is generally stronger when eddies have higher intensity and is confirmed by altimetry observation that eddies with greater amplitudes exhibit stronger axisymmetry (Supplementary Fig. S4). Given that mesoscale eddies are non-linear coherent structures in the rotating and stratified fluid, the Rossby number $Ro = U/fR$ is introduced as a dynamical index to describe the intensity of eddies, where $f$ is the Coriolis parameter and $R$ is the radius of the eddy. As shown in Fig. 2d and consistent with dynamical expectations, eddies tend to be more axisymmetric with larger Rossby numbers.

The elongated eddies have a directionally-dependent nature which can be represented by the azimuth angle $\theta$ of their major axis as shown in Fig. 1c. The distribution of eddy direction is not random but has a regular pattern. As illustrated by the ellipses in Fig. 3a, eddies in high latitudes tend to have a more meridional direction, and they tend to be more zonal in low latitudes, consistent with former observational

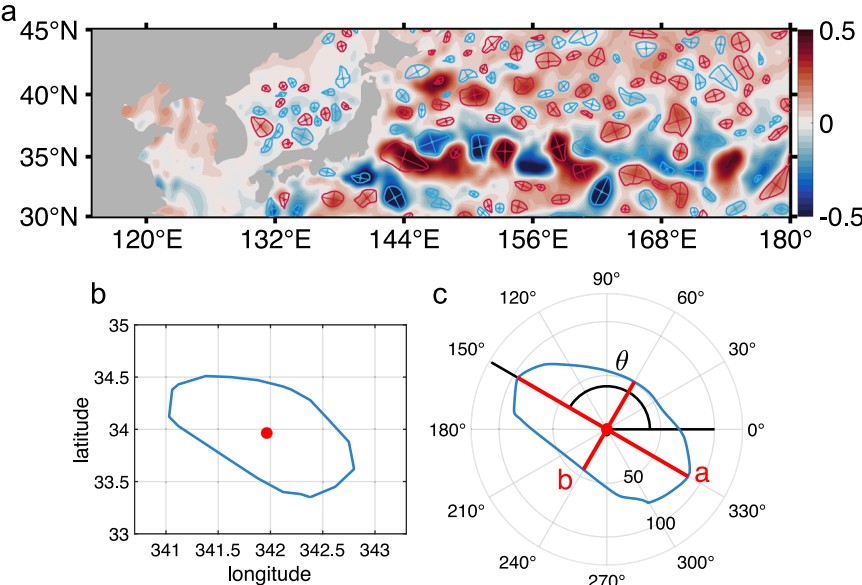

**Fig. 1 | Asymmetry and directional-dependence of oceanic mesoscale eddies.** **a** A snapshot of sea surface height anomaly (SSHA) distribution in Kuroshio Extension region on May 16, 2015 and the corresponding eddies' major and minor axes. Red curve represents the edge of anti-cyclonic eddies, and blue curve represents the edge of cyclonic eddies. Crossed lines within an eddy denote the major and minor axis, respectively. Color represents the SSHA in unit meter. **b** A typical eddy in the geographic coordinate. Red spot is the eddy center and the closed blue curve represents the eddy edge. **c** The same eddy in the eddy-centric polar coordinate. Long red line is defined as the major axis $a$ and the short one is defined as the minor axis $b$. The angle between the major axis and the positive x-axis is defined as the eddy direction angle $\theta$. Source data are provided as a Source Data file.

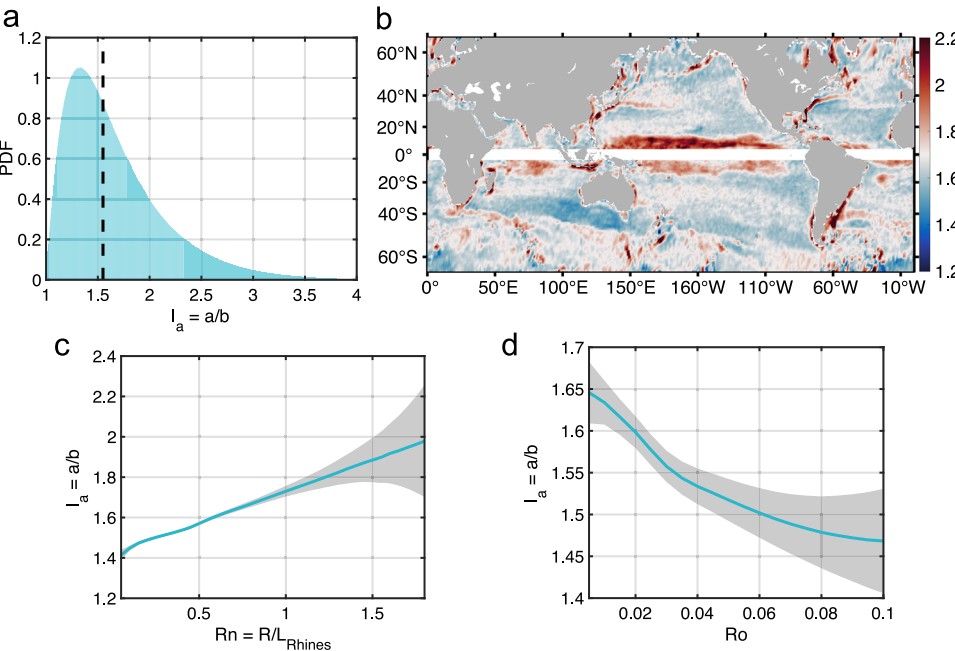

**Fig. 2 | Non-axisymmetry index $I_a$ of mesoscale eddies and its determinant factors. a** The probability density function (PDF) histogram of non-axisymmetry index defined by $I_a = a/b$. The x-axis represents $I_a$ and the y-axis is the corresponding probability density function as a function of $I_a$. The vertical dash line represents the average value of $I_a$, which is 1.55. **b** Global distribution of the non-axisymmetry index $I_a$ constructed by averaging with a $1° \times 1°$ moving window. **c** Globally-averaged curve of index $I_a$ as a function of normalized eddy radius by Rhines Scale ($R_n = R/L_R$). The data is removed in the area near the equator where the geostrophic balance does not hold. **d** Globally-averaged curve of index $I_a$ as a function of Rossby number ($Ro$) of eddies. Blue curve in each subfigure represents the average value and gray shading represents the error bar computed by the standard error of average. Source data are provided as a Source Data file.

results[47,48]. In order to evaluate eddies' directional-dependence, $I_d = \sin(\theta)$ is introduced as an directional index, where the azimuth angle $\theta$ ranges from 0 to $\pi$. Thus defined, meridionally directed eddies have $I_d$ close to 1 and zonally directed eddies have $I_d$ close to 0. The global distribution of $I_d$ (Fig. 3b) shows that eddies in high latitudes are meridionally directed and, in low latitudes, they are more zonal.

The observed distribution of $I_d$ in Fig. 3b also exhibits some fine structures. In the tropics, Kuroshio, and Gulf Stream regions, eddies are influenced by the large-scale circulation and have a tendency of zonal directional-dependence. Additionally, topography also influences the eddy direction. The eddies near mid-ocean ridges and along continental boundaries tend to be parallel to the local topographic

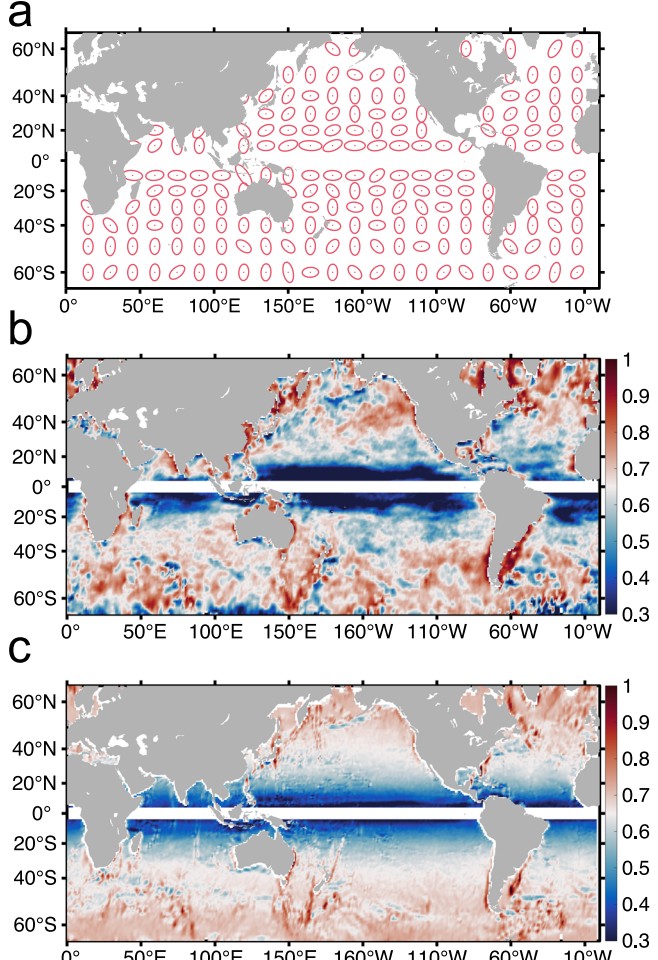

**Fig. 3 | Global distribution of the observed and reconstructed eddy directional index $I_d$. a** Map of eddy direction expressed in ellipses. Eddy direction angle is averaged with a $5° \times 5°$ grid and a $5°$ window size. **b** Global distribution of the observed eddy directional index $I_d = \sin(\theta)$, where $\theta$ is the eddy direction angle. This map is constructed by averaging with 1° moving window at each grid point for all available data points. **c** The reconstructed global distribution of index $I_d$ by a fitting function of local planetary $\beta$, geostrophic velocity and bottom topography gradient (see Methods for details). The correlation coefficient of the fitted $I_d$ with the observed value is 0.73, and the relative error of the reconstruction is 36.5%. The data is removed in the area near the equator where the geostrophic balance does not hold. Source data are provided as a Source Data file.

contours. Considering the gradient of PV caused by stratification can be represented approximately by the surface large-scale geostrophic velocity[57,59,60] and the topographic slope also can serve as a topographic PV gradient[61], all the background influencing factors can be unified within the concept of PV gradient.

By combining the planetary, stratification and topography $\beta$ effect, it is possible to reconstruct the global distribution of $I_d$ with a reasonable accuracy (see Method for details). The correlation coefficient of the fitted $I_d$ with the observed value is 0.73, and the relative error of reconstruction is 36.5%, both of which pass the bootstrap test (Supplementary Table S1). This means that all the factors considered here have their respective contributions.

### Heat flux influenced by eddy directional-dependence

As shown in Fig. 4, mesoscale eddies with different directions exhibit distinct variations in their SSHA distribution and velocity field. Even with identical background temperature fields, the corresponding eddy-induced heat fluxes are expected to be quite different. For convenience, all available eddy observations are divided

into four directional groups: meridional, zonal, northeast and southwest (NESW), and northwest and southeast (NWSE). Each category occupies a specific range of azimuth angle $\theta$ as shown in Fig. 4a–d. Among all observed eddies, the meridional directed eddies consist of 29.2%; and the percentage for zonal, NESW and NWSE directed eddies are 28.6%, 21.5% and 20.7%, respectively. The composite SSHA distributions of these four types of differently directed eddies all exhibit obvious elongation along their major axes as shown in Fig. 4a–d.

The directional distributions of eddy-induced surface heat flux **Q** for the four types of eddies in the Northern Hemisphere are shown in Fig. 4e–h. Although all the eddy-induced surface heat fluxes exhibit a northward tendency resulting from the background meridional temperature gradient of Northern Hemisphere, the heat flux directions for each category are different and are determined by the corresponding eddy directional-dependence. On average, the zonally directed eddies induce a nearly eastward heat flux, the meridionally directed eddies a nearly northward heat flux, the NESW directed eddies a nearly northeastward heat flux, and the NWSE directed eddies a nearly northwestward heat flux. Similar results are also observed in the Southern Hemisphere, except that the eddy heat flux has a general southward tendency (Supplementary Figs. S7, S8).

The mechanism of eddy direction influencing the eddy heat flux is a combined effect of sea surface temperature anomaly (SSTA) and surface velocity anomaly fields, as demonstrated by the composite results of satellite observations in Fig. 4i–l. Because the largest gradient of SSHA is along the minor axis direction, the strongest velocities emerge in the major axis direction. For the meridionally directed eddies in the Northern Hemisphere, their meridional velocity field reaches its largest extremes in the west/east of the eddy center, advects effectively the background meridional temperature gradient, and induces a strong temperature anomaly dipole. As shown in Fig. 4i, strong meridional velocity and large temperature anomaly induces a dominant meridional heat flux. In contrast, the zonally directed eddies have weak meridional velocity in the west/east of the eddy center, leading to a weak temperature anomaly and a small meridional heat flux, as shown in Fig. 4j and the dominant direction of the heat flux is zonally directed. Therefore, all averaged directions of eddy-induced heat fluxes are generally parallel to the eddy's major axes of the corresponding category as shown in Fig. 4e–h. The similar results are also found by compositing the surface drifter data, which indicates that this mechanism is valid irrespective of the assumption of geostrophic balance (Supplementary Fig. S9).

Although, the percentages of the four types of differently directed eddies are quite close (20–30%), the meridionally directed eddies can induce meridional heat flux most effectively and can make the most important contribution to the total meridional heat flux. The global distributions of the meridional heat flux induced by the four types of differently directed eddies share similar global patterns, but their amplitudes deviate significantly, as shown in Fig. 5a–d. The meridional directed eddies have a global average value of $2.42 \times 10^{-3}$ m s$^{-1}$ °C, which doubles the global average eddy heat flux by the zonally directed eddies of $1.18 \times 10^{-3}$ m s$^{-1}$ °C. The NESW and NWSE directed eddies have global average values in between at $1.60 \times 10^{-3}$ m s$^{-1}$ °C and $1.23 \times 10^{-3}$ m s$^{-1}$ °C, respectively. As shown in Fig. 5e, the domination of the heat flux induced by meridionally directed eddies is also demonstrated by the average heat fluxes in western boundary current and ACC regions, where the strongest meridional heat flux emerges. The zonally averaged meridional heat fluxes are further computed for the four eddy types, as shown in Fig. 5f. The meridional distributions of the zonally averaged heat fluxes have positive peaks near $35°N - 40°N$ and negative peaks near $30°S - 40°S$, which are in general consistency with patterns in former estimates[14–17].

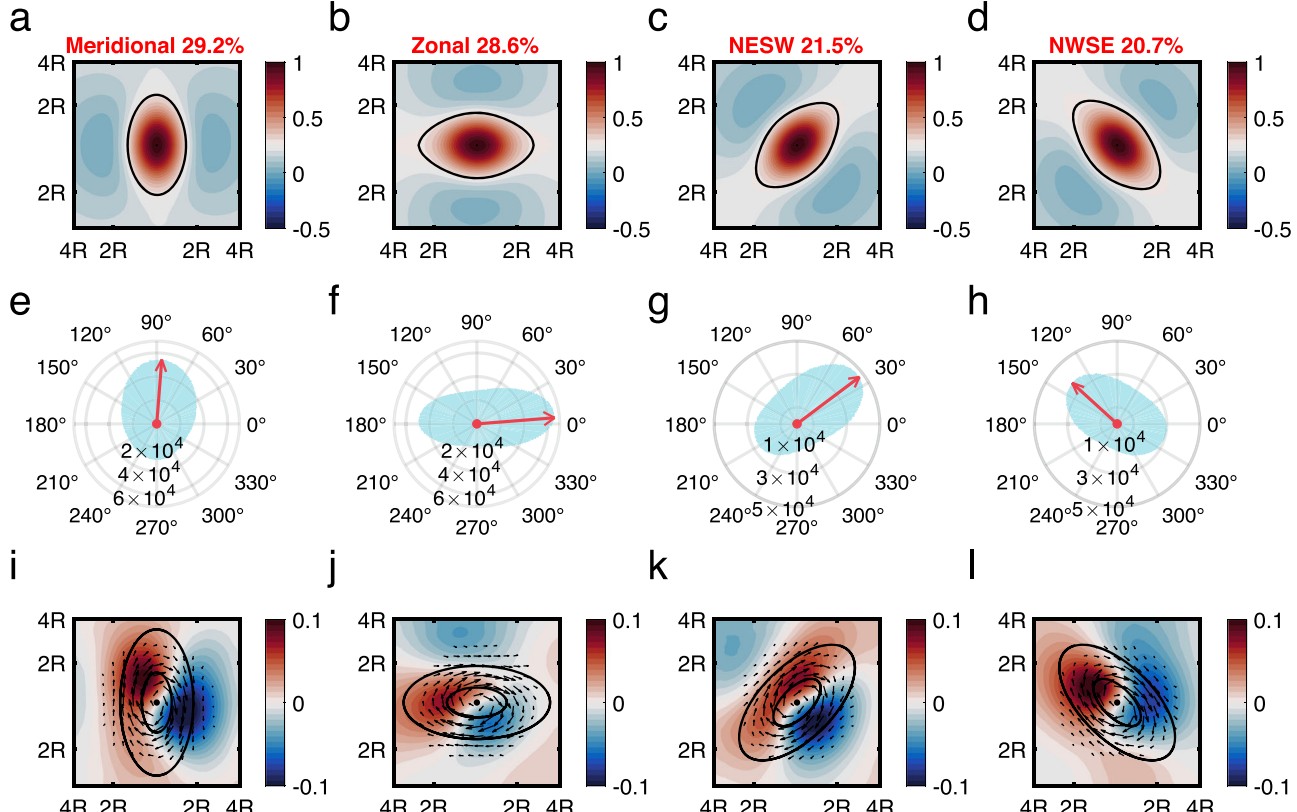

**Fig. 4 | Direction of eddy-induced heat flux controlled by eddy directional-dependence.** Spatial structures of composited normalized sea surface height anomaly (SSHA) distributions in the normalized eddy-centric coordinate of **a** meridional, **b** zonal, **c** Northeast-Southwest (NESW) and **d** Northwest-Southeast (NWSE) directed eddies, respectively. The azimuth angle $\theta$ ranges are $(3\pi/8, 5\pi/8)$ for meridional directed eddies; $(0, \pi/8)$ and $(7\pi/8, \pi)$ for zonal directed eddies, $(\pi/8, 3\pi/8)$ for NESW directed eddies, and $(5\pi/8, 7\pi/8)$ for NWSE directed eddies. Color represents the normalized SSHA and black bold contours represent eddies' boundary. Red number over each subfigure represents the percentage of eddies with the corresponding direction among all observed eddies. Eddy-induced surface heat flux directions by the **e** meridional, **f** zonal, **g** NESW and **h** NWSE directed

eddies in the Northern Hemisphere. Blue shade is the eddy counting numbers for each direction and red vector represents the averaged direction of the eddy-induced heat flux of all available eddies with the corresponding direction. Panels **i**–**l** are the corresponding sea surface temperature and surface velocity anomaly fields of anticyclonic eddies in the Northern Hemisphere, both composited by satellite remote sensing data. Black vectors represent the surface velocity anomaly field, and color shade represents the sea surface temperature anomalies (SSTA). Only the asymmetric part of the temperature anomaly is shown here for demonstration, which is computed by subtract the symmetric SSTA $T_0(\boldsymbol{r})$ from the total SSTA after composition. Black contours in each subfigure represent the ideal eddy boundary. Source data are provided as a Source Data file.

## Heat flux influenced by eddy asymmetry

The result of eddy's directional-dependence in influencing the direction of eddy-induced heat flux suggests the asymmetric motions of eddies should have a significant impact on heat flux. To quantify this impact, we decompose the eddy velocity field into symmetric and asymmetric components. Take the meridionally elongated anticyclonic eddy as an example, its symmetric velocity component corresponds to the symmetric circular SSHA field, as shown in Fig. 6c. The asymmetric SSHA is the difference between the original SSHA and the symmetric part, exhibiting a four-petal structure with positive anomalies along the major axis and negative ones along the minor axis, as shown in Fig. 6b. The corresponding asymmetric velocity field is featured by two curved meridional jets, including a northward jet on the left and a southward jet on the right. These two jests traverse the temperature anomaly dipoles in Fig. 4i, which suggests that they can induce meridional heat flux quite effectively.

In order to quantify the relative amplitude of the symmetric and asymmetric components of the eddy velocity field, eddy kinetic energy (EKE) is also decomposed into symmetric and asymmetric parts (see Method for details). With the decomposition adopted here, the sum of the symmetric and asymmetric parts is exactly equal to the total EKE. As shown in Fig. 6d, e, the global distributions of symmetric and asymmetric EKE have similar patterns, both of which are featured by enhanced EKE along the western boundary currents and the ACC. The

global average asymmetric EKE is $8.7 \times 10^{-3}$ m² s⁻², which is very close to the average value of its symmetric counterpart at $9.3 \times 10^{-3}$ m² s⁻². This nearly equal partition is also confirmed by the zonally averaged EKE curves in Fig. 6f.

It is naturally expected that the eddy induced heat flux will be substantially influenced by this EKE partition. By keeping the temperature distribution unchanged, the heat fluxes induced by the symmetric and asymmetric velocity components are computed. As shown in Fig. 7a, b, the meridional heat fluxes induced by the symmetric and asymmetric velocity components share similar global distributions and amplitudes. The global average meridional heat fluxes of the asymmetric and symmetric parts are $2.8 \times 10^{-3}$ m s⁻¹ °C and $3.6 \times 10^{-3}$ m s⁻¹ °C, consisting of 44% and 56% of the total EHF, respectively. Meanwhile, in some regions with strong meridional EHF, such as the Leeuwin Current along the west coast of Australia and the Agulhas Current, the heat fluxes induced by asymmetric parts can slightly exceed the symmetric counterparts (Fig. 7c). The zonally averaged curves of the meridional heat fluxes induced by the symmetric and asymmetric velocity components also show a nearly equal partition as shown in Fig. 7d.

The eddy-induced heat transport can also be decomposed into trapping and stirring components by partitioning the temperature anomaly field into symmetric and asymmetric parts[16,50,51]. Since the main focus of our work is on the asymmetry of the eddy's dynamical

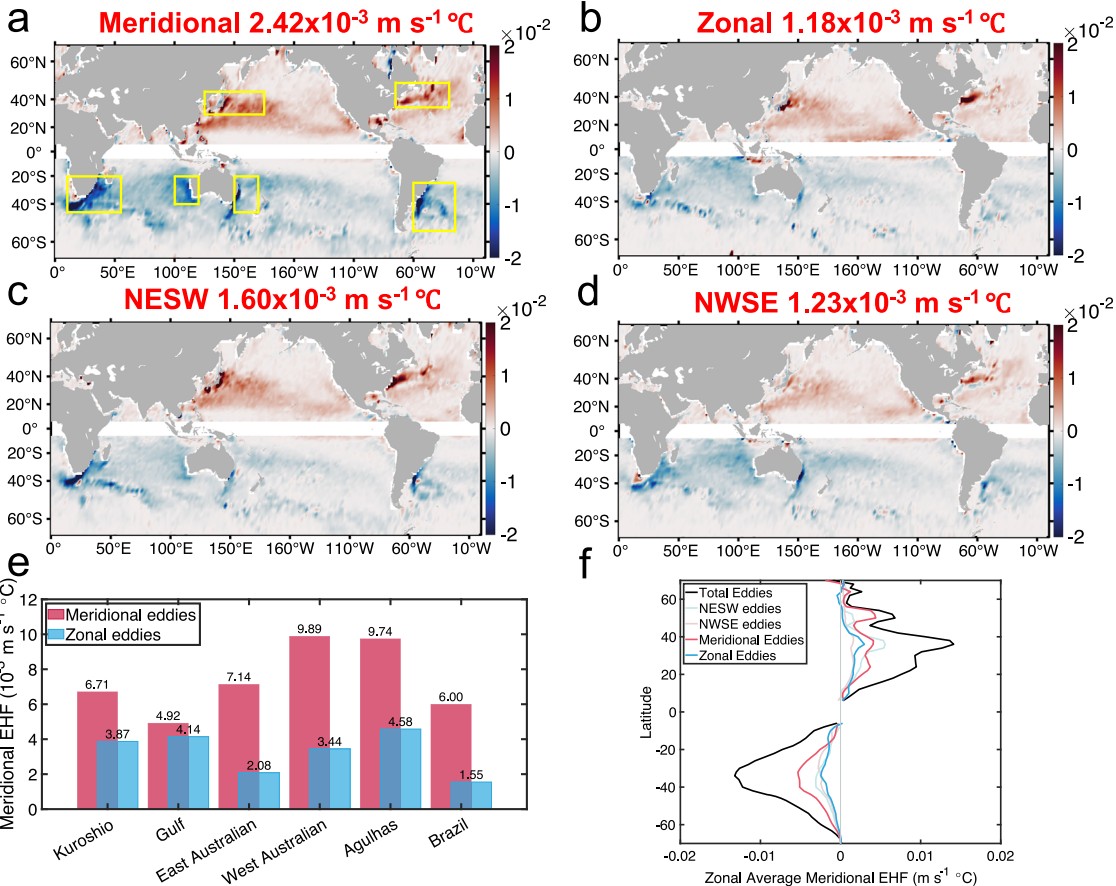

**Fig. 5 | Global distributions of meridional eddy heat flux induced by eddies with different directions. a–d** Global distributions of averaged meridional heat flux induced by four types of eddies with different directions. **a** Meridional, **b** zonal, **c** Northeast-Southwest (NESW) and **d** Northwest-Southeast (NWSE), which are constructed by averaging with a $2° \times 2°$ moving window. Yellow boxes in (**a**) mark the regions with strong eddy heat fluxes. Red numbers above each subfigure are the globally averaged eddy-induced heat flux for each type of directed eddies, which are computed after multiplying the heat flux in the southern hemisphere with −1. The data is removed in the area near the equator where the geostrophic balance does not hold. **e** Averaged meridional heat fluxes in the regions indicated within the yellow boxes in **a**. Red bars indicate the values of meridional directed eddies and blue bars for zonal ones. Numbers over the bars represent the average meridional heat fluxes in the boxes with unit $10^{-3}$ m s$^{-1}$ °C. **f** Zonally averaged meridional heat fluxes induced by eddies with different directions. The sum of eddies' zonally integrated meridional heat fluxes is represented by the black curve. Source data are provided as a Source Data file.

structure (velocity and pressure field) and its influence on the heat transports, we do not separate the temperature anomaly field. Thus, the **Q** defined in our study is the total eddy induced surface heat flux, containing both the stirring and the trapping effects.

## Discussion

By combining the satellite remote-sensing and surface drifter data, we show the degree of asymmetry of oceanic mesoscale eddies and reveal its contributing dynamical factors. The direction and amplitude of eddy-induced heat flux are substantially influenced by the asymmetry and directional-dependence of eddies. If the eddy velocity field is decomposed into asymmetric and symmetric parts, the corresponding EKEs exhibit a nearly equal partitioning by these two velocity components. The total eddy induced meridional heat fluxes almost double the heat flux induced only by the symmetric velocity components. The magnitude of the total eddy induced meridional heat fluxes also increase with the eddy asymmetry on a global average sense. In mid-high latitude regions, the asymmetry can increase the total eddy-induced meridional heat flux by about 50% or even double the heat flux (Supplementary Figs. S1 and S2), which highlights the crucial contribution of eddy asymmetry on heat transports.

Since the mesoscale eddies are typically not included in the present-day coarse resolution climate models, parameterizations of

eddy effect are widely used by these models. Our results here show the transports by eddies are far from simple down-gradient diffusion. This can potentially introduce large relative errors up to 50–100% when computing the heat flux terms in the climate models. Our results also show that the asymmetry and directional-dependence of eddies have specific global patterns, and quantitative relations are established to describe the influences of large-scale factors. These results could provide useful observational constrain for improvements of future parameterizations in the coarse resolution climate models. For the eddy-resolving high resolution climate models, our results provide an observational baseline to test the veracity of these models, especially, an asymmetry perspective is added for verifying the simulation of mesoscale eddies.

Since the eddy-induced transports can also have an important influence on the large-scale circulation through eddy-mean flow interaction and inverse energy cascade[62,63], a key question arising naturally is whether the relation between eddy asymmetry and large-scale factors can establish climatic feedback. Considering that the mesoscale eddies can be a major contributor to the interannual to decadal variation of basin-integrated meridional heat fluxes[19], there is a possibility that the feedback between eddy asymmetry and large-scale motions can serve as a potentially important mechanism in driving long-term climate variability or oscillation. Future studies are needed in both establishing and quantifying this potential feedback.

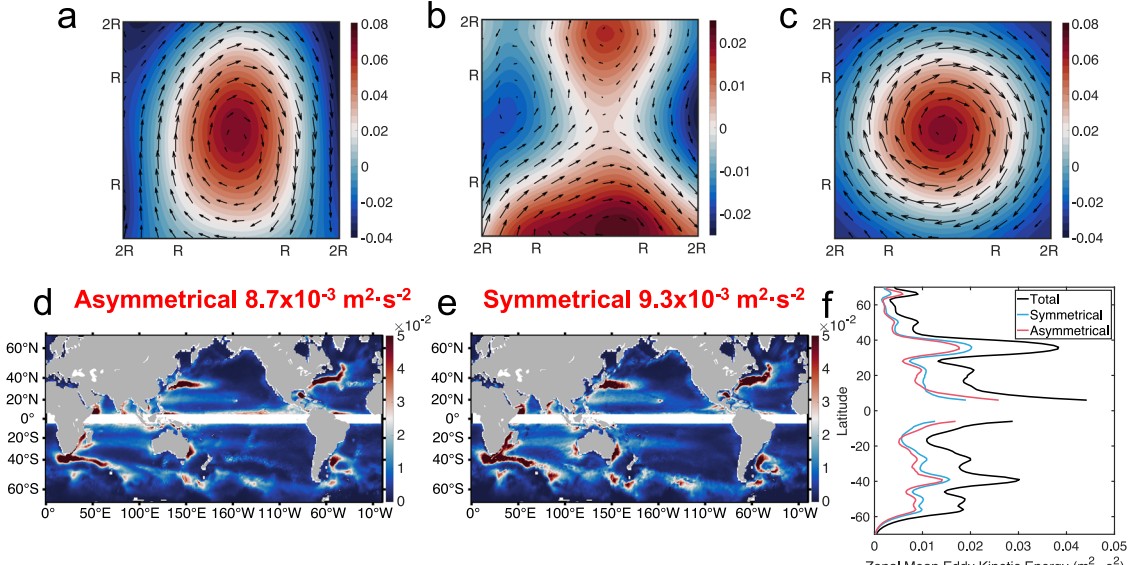

**Fig. 6 | Global distribution of eddy kinetic energy (EKE) of asymmetric and symmetric parts of the eddy flow field. a** Snapshot of the observed sea surface height anomaly (SSHA) $\eta$ and geostrophic velocity anomaly of an elongated eddy observed on January 3, 1993 and at 161.48°E, 48.56°N. Color represents the SSHA and vectors represent the flow field. **b** Asymmetric part of the SSHA ($\eta_a$) and geostrophic velocity anomaly ($V'_a$). **c** Symmetric part of the SSHA ($\eta_s$) and geostrophic velocity anomaly ($V'_s$). The observed SSHA is decomposed into the asymmetric/asymmetric parts as $\eta = \eta_a + \eta_s$, and the corresponding velocity fields are computed by geostrophic relation. **d** Global distribution of the averaged asymmetric EKE, which is constructed by averaging with a 1° ×1° moving window. The global mean asymmetric EKE is $0.0087 m^2 \cdot s^{-2}$. **e** Global distribution of the averaged symmetric EKE. The global mean symmetric EKE is $0.0093 m^2 \cdot s^{-2}$. **f** Zonal mean EKE of total (black curve), symmetric (blue curve), and asymmetric component (red curve) as a function of latitude. The data is removed in the area near the equator where the geostrophic balance does not hold. Source data are provided as a Source Data file.

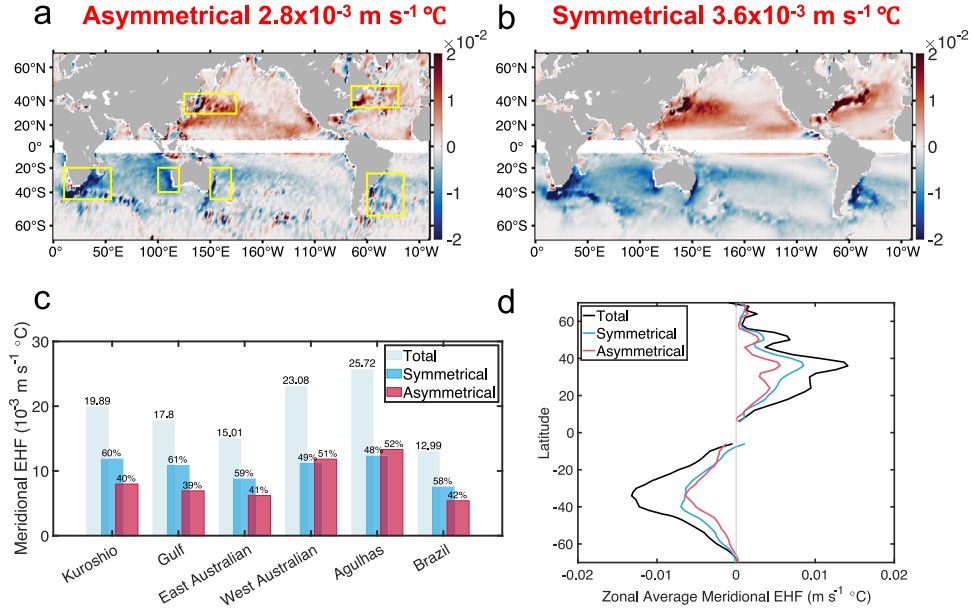

**Fig. 7 | Global distributions of the symmetric and asymmetric meridional eddy induced heat fluxes.** Panel **a** represents the averaged meridional heat flux induced by the asymmetric eddy flow field, which is constructed by averaging with a 2° ×2° moving window. Yellow boxes in (**a**) mark the regions with strong eddy heat flux. Panel **b** represents the symmetric parts. **c** Total, symmetric and asymmetric part of meridional heat fluxes in the regions indicated within the yellow boxes in **a**. Light bars indicate the values of total meridional heat flux in these boxes, and numbers over the bars represent the average meridional heat fluxes in the boxes with unit $10^{-3}$ m s$^{-1}$ °C. Blue and red bars indicate the values of symmetric and asymmetric part of the meridional heat fluxes, respectively, and the numbers over the bars represent the percentage over the total flux. **d** Zonally averaged meridional heat fluxes of total (black curve), symmetric (blue curve) and asymmetric part (red curve) as a function of latitude. The data is removed in the area near the equator where the geostrophic balance does not hold. Source data are provided as a Source Data file.

## Methods

### Satellite datasets

The Global Ocean Gridded L4 Sea Surface Heights and Derived Variables Reprocessed 1993 Ongoing product provided by CMEMS are used here. This multiple-satellite-merged data contains global gridded daily SSHA $\eta$ and geostrophic velocity anomaly $\mathbf{V'}$ fields with a ¼ degree resolution from 1993 to 2021. The NOAA OI SST V2 High Resolution Dataset is used to estimate the eddy heat flux, which has the same resolution and time coverage as the SSHA data product above.

### Drifter dataset

The drifter data used here is provided by the Drifter Data Assembly Center (DAC) of National Oceanic and Atmospheric Administration (NOAA). The DAC assembles and provides uniform quality control of sea surface temperature (SST) and surface velocity by satellite-tracked surface drifting buoy observations. The time resolution of the drifter data is about 6 hours and the time span of the drifter data is from 1998 to 2017, which have a global coverage and contain totally 33,960,935 observational data points.

### Eddy tracking data

This study's mesoscale eddy dataset is the Mesoscale Eddy Trajectory Atlas Product (META ver3.2) provided by AVSIO. The mesoscale eddies in this dataset are detected from the Absolute Dynamic Topography (ADT) field. We use the all-satellites version, which gives the daily longitude, latitude, eddy edge contours, amplitude, radius $R$, and amplitude of eddy rotational speed $U$ with time range from 1993 to 2021. It contains 35,939,078 cyclonic and 34,206,331 anticyclonic instances of eddy identification (with eddy life span longer than 10 days).

### Topography data

The Topography Data used here is the ETOPO1 One Arc-Minute Global Relief Model data provided by the National Oceanic and Atmospheric Administration (NOAA). It is vertically referenced to sea level, and horizontally referenced to the World Geodetic System of 1984 (WGS 84). The grid resolution of ETOPO1 is 1 arc-minute.

### Major and minor axes

The latitude/longitude of eddy center and eddy edge curve are given by the AVISO eddy dataset. Establish a polar coordinate system with the eddy center as the pole. This polar coordinate is called the eddy-centric coordinate. The original latitude and longitude of eddy edge under geographic coordinate are transformed into the eddy-centric coordinate. The distance and azimuth between eddy edge and eddy center is $r$ and $\theta$ in the eddy-centric coordinate. Under the eddy-centric coordinate, the longest line across the eddy center is defined as the major axis $a$. The line perpendicular to the major axis is defined as the minor axis $b$, as shown in Fig. 1c.

### Reconstruction of directional index

The zonal average directional index $I_d$ could be fitted by the planetary beta:

$$\bar{I}_d = f(\beta) = 0.7598 - 0.1142\beta - 1.3558e^{19.0726\beta - 20.7233} \tag{1}$$

where $\beta = \cos(latitude)$, which has been normalized by its maximum value at equator.

The stratification $\beta$ effect is introduced as $\boldsymbol{\beta}_{\mathbf{g}} = (\bar{u}_g, \bar{v}_g)$, where $\bar{u}_g$ and $\bar{v}_g$ are the zonal and meridional muti-year average geostrophic current given by the altimetry data. The PV gradient caused by the ocean bottom topography is introduced as $\boldsymbol{\beta}_{\mathbf{b}} = (H_{bx}, H_{by})$, where $H_{bx}$ and $H_{by}$ are the zonal and meridional gradient of the ocean water depth $H_b$.

Combining all the factors mentioned above, a simple linear fit is applied to construct the global map of directional index $I_d$ as:

$$I_d = x_0 + x_1 \cdot f(\beta) + \mathbf{x_2} \cdot \boldsymbol{\beta}_{\mathbf{b}} + \mathbf{x_3} \cdot \boldsymbol{\beta}_{\mathbf{g}} \tag{2}$$

where $f(\beta)$ is the regression function in Eq. (1); $x_0 \sim x_3$ are the fitting parameters.

### Eddy-induced heat flux

The surface eddy-induced heat flux (EHF) is defined as:

$$\mathbf{Q} = \overline{\mathbf{V'}T'} = \left(\overline{u'T'}, \overline{v'T'}\right) \tag{3}$$

where $u'$ and $v'$ are the zonal and meridional geostrophic current anomalies given by altimeter data; $T'$ is the temperature anomaly given by the instantaneous satellite observations of sea surface temperature minus the climatological monthly average temperature from 1993 to 2019. The overbar in (3) denotes average within a square with the size ±2 eddy radii around the eddy center. Based on the daily satellite altimetry and remote sensing data, each eddy has a corresponding snapshot of the surface geostrophic current anomaly and temperature anomaly. Thus, we have a surface heat flux vector $\mathbf{Q}$ induced by each eddy given by Eq. (3).

### Eddy structure composition

Composite SSHA distributions of eddies with four types of different directions in Fig. 4a–d are computed in four steps: First, the SSHA data of each eddy is projected to the eddy-centric coordinate $(x_c, y_c)$. Second, the SSHA map of eddy is normalized by its local extreme value at the eddy center for both cyclonic and anticyclonic eddies; Third, the distance to the eddy center is normalized with eddy radius $(x_n, y_n) = (x_c, y_c)/R$; Finally, the normalized SSHA maps are averaged in the normalized eddy-centric coordinate $(x_n, y_n)$ for composition of all eddies with the same direction.

The composite distributions of SSTA and surface geostrophic velocity anomaly in Fig. 4a–d are computed in two steps: First, the SSTA and velocity of each eddy is projected to the normalized eddy-centric coordinate $(x_n, y_n)$; Second, average the SSTA and velocity in $(x_n, y_n)$ for composition of all eddies with the same direction. Since the axisymmetric part of the temperature anomaly does not contribute significantly to the heat flux, subtract the symmetric SSTA $T_0(r)$ from the composite SSTA after composition, where the symmetric SSTA $T_0(r)$ is computed through averaging the SSTA along the tangential direction in the eddy-centric polar coordinate.

### Eddy velocity field decomposition

The eddy-induced flow field is divided into symmetric and asymmetric components as follows. For each eddy, the SSHA $\eta(x_c, y_c)$ in the eddy-centric coordinate is decomposed into symmetric and asymmetric parts. The symmetric SSHA $\eta_s(r)$ is computed through averaging the SSHA along the tangential direction in the eddy-centric polar coordinate. Subtract the symmetrical SSHA $\eta_s$ from the original SSHA $\eta$ to get the asymmetric part $\eta_a$, as shown in Fig. 6. Thus, the SSHA of an eddy is linearly decomposed into symmetric and asymmetric part:

$$\eta(x_c, y_c) = \eta_a(x_c, y_c) + \eta_s(r) \tag{4}$$

Using the symmetric and asymmetric SSHA, the corresponding geostrophic velocity anomalies can be calculated based on the geostrophic balance. As shown in Fig. 6, the total geostrophic velocity field $\mathbf{V'}$ of the eddy is linearly decomposed into a symmetric part $\mathbf{V'_s}$ and an asymmetric part $\mathbf{V'_a}$. These velocity fields satisfy a simple linear relation: $\mathbf{V'} = \mathbf{V'_a} + \mathbf{V'_s}$. Based on the velocity decomposition, EKE could also

be divided into symmetric and asymmetric part:

$$\text{EKE} = \text{EKE}_a + \text{EKE}_s = \frac{1}{2}\mathbf{V_a'}^2 + \frac{1}{2}\mathbf{V_s'}^2 \tag{5}$$

The cross-product term $\mathbf{V_a'} \cdot \mathbf{V_s'}$ is zero theoretically. For the heat flux of each eddy, the linear partition also holds. Thus, the eddy-induced heat flux can be linearly decomposed into two parts:

$$\overline{\mathbf{V'}T'} = \overline{\mathbf{V_a'}T'} + \overline{\mathbf{V_s'}T'} \tag{6}$$

## Data availability
The altimeter SSHA and geostrophic velocity data can be accessed from: https://data.marine.copernicus.eu/product/SEALEVEL_GLO_PHY_L4_MY_008_047/description. The sea surface temperature data can be accessed from: https://psl.noaa.gov/data/gridded/data.noaa.oisst.v2.highres.html. The surface drifter dataset can be downloaded from: ftp://ftp.aoml.noaa.gov/phod/pub/buoydata. The global mesoscale eddy trajectory data be accessed from the: https://www.aviso.altimetry.fr/en/data/products/value-added-products/global-mesoscale-eddy-trajectory-product.html. The topography data can be accessed from: https://www.ncei.noaa.gov/metadata/geoportal/rest/metadata/item/gov.noaa.ngdc.mgg.dem%3A316/html. Source data are provided with this paper.

## Code availability
Matlab 2022 was used to plot the figures.

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

## Acknowledgements
This research was supported by the National Natural Science Foundation of China under Grants 42022041, 42288101 and 41876001 of Z.Z.

## Author contributions
H.W., B.Q. and Z.Z. contributed to design of the study, data processing and analysis, interpretation of the results and writing of the manuscript. H.L. contributed to discussing the results and improving of the manuscript.

## Competing interests
The authors declare no competing interests.

## Additional information

**Peer review information** *Nature Communications* thanks Andrew Del-man, Tongya Liu, and the other, anonymous, reviewer(s) for their con-tribution to the peer review of this work. A peer review file is available.

