## [Peer review file · Nature Communications]

REVIEWER COMMENTS

Reviewer #1 (Remarks to the Author):

By processing the available data based on satellite altimetry (geostrophic velocities and sea surface height anomalies) and the data on sea surface temperature the manuscript analyses the deviations of the ocean mesoscale eddies from the circular shape and shows that the elongated form of eddies is essential in defining the eddy induced mean eddy heat flux.

Although these results are of interest, I cannot recommend the manuscript in present form as some of its statements are misleading and require major revision. The authors build their motivation on the argument that the existing eddy parameterizations rely on the assumption that eddies are of circular shape, which is not the case. They mix the notion of axisymmetric eddy with that of isotropic turbulence. The isotropy in the latter case is understood in statistical sense and does not exclude eddies of arbitrary shape. Numerous eddy simulations show the presence of elongated structures in the vorticity field even in isotropic quasi-two dimensional turbulence. Furthermore, asymmetry of eddies translates in effective eddy mixing (which incorporates the effect of multiple random eddies) in a very indirect way. Of course, in many cases modelers use isotropic diffusivities in downgradient parameterizations. However, this is done for simplicity reasons. There is understanding than mixing is not isotropic, and, for example, Smith and Gent (J.Phys. Oceanogr. 34, 2541--2564, 2004) discuss a mathematical framework to use anisotropic diffusivities.

The authors see their finding in the demonstration that the mesoscale eddies in the ocean do not have an axisymmetric form. However, there were no claims in the literature that they are. Ocean mesoscale eddies are generated through baroclinic and barotropic instabilities which occur in the presence of gradients of potential vorticity. This immediately makes the direction along the gradient and cross the gradient non-equivalent.

The partition of eddy velocity field into axisymmetric and the rest illustrate that the deviations from symmetry are not negligible. However, the value of the exercise with eddy heat flux is not immediately clear to me (and it determines the title of the manuscript!). If one takes only a part of real velocity one gets a smaller heat flux. The doubling mentioned in the title of the manuscript is only the consequence of taking ARTIFICIALLY a part of perturbation. Heat flux is determined by correlations. The shape of velocity and temperature perturbations is the result of evolving instabilities which release available potential energy and hence are accompanied by heat flux. Why it is of interest to decompose real eddies into symmetric and antisymmetric parts? The symmetric part taken separately is not a solution of dynamical equations. One needs the entire perturbations. There is no use in comparing with non-existing solutions.

In summary, the information presented in the manuscript might be of interest, but motivation of the manuscript is questionable.

Minor points:

line 10 or axisymmetric vortex? --- What do you mean?

12 are asymmetric and orientative?? --- Do you simply mean that they are elongated? If yes, just say so.

31 redistribution

36 of mass?

44-47 But simulations generally show large spread, and I do not think that at present the amount of high-resolution simulations is sufficient to do reliable conclusions.

53 '...this calls for a urgent need...' ? --- Edit

63 '...density variations cause tilted isopycnals ...' --- What do you mean?

65-66 I do not think this is a correct statement: The effect of eddies is parameterized as a product of thickness diffusivity with the slope gradient. The latter takes into account asymmetry. As concerns the diffusivity, it is sometimes taken as a constant, but sometimes is a subject of separate parameterizations. If it is taken as a constant, it is not because we think that eddies are circular, this is just the simplest choice. There are more sophisticated choices that try to work with local balances.

70-71 Nobody has doubts in this. Nobody is really relying on isotropic turbulence

72 curvature of the Earth is not the same as beta-effect

73 'spontaneously' --- What do you mean?

73-75 Nobody thinks differently, why do you try to prove this?

77 Which approximation?

78-79 Downgradient diffusion and axisymmetry of vortices are not related generally. Asymmetric vortices may lead to downgradient diffusion, and axisymmetric vortices will not necessarily result in downgradient diffusion.

83, 84 Again, this is wrong statement: Eddies always exist of the background and even the onset of instability is related to the structure of PV gradient. It is the presence of background gradients that makes eddies as they are.

88-90 Taken generally, this is a trivial statement, one would be surprised if it were otherwise. Of course the magnitude of the effect is of interest.

92-93 This statement should be corrected: You split velocity into two parts, and there is nothing surprising in what you see. Your symmetric velocity corresponds to smaller energy and smaller heat flux. So what is claimed? That part is smaller than the whole? Eddies are as they are created by dynamics, there is no need in replacing them by symmetric eddies.

102, 103 f-plane is not what is written here

105-106 Only in conditions of weak background gradients.

106-107 Freely evolving turbulence has very little to do with oceanic turbulence (and please use quasi-geostrophic)

108-109 May be in simple theories, but not otherwise.

132 '...rotational effect of stratified fluid is not homogenous due to the Earth's surface curvature' --- please provide a correct definition of PV. Why are you talking about Earth's curvature? Do you mean

Earth sphericity? Or simply the latitudinal dependence of the Coriolis parameter?

166-167 Do you mean planetary PV gradient? In reality PV gradient depends on vertical velocity shear and the gradient of relative vorticity of background flow. Why everything should depend on planetary PV gradient? It is not necessarily dominant in many places. The Rhines scale is obviously related to flat-bottom barotropic turbulence, and its appearance here is questionable at least.

182-183 This is a recurrent phrase, but the ability of an eddy to preserve its shape depends on background state, as I have already mentioned.

191 "The elongated eddies have an orientative nature" ? Are you willing to say that elongated eddies can be characterized by the direction of major axis to zonal direction? If yes, then say

193 has

210 PV gradient is not only beta-effect.

224 'with high density of crowded eddies' -- with high density of eddies

226-227 But why? If there are many eddies, the strain created by them will be randomly distributed.

234-236 It is a repetitive statement which contains no information without writing formulas. Eddies are always in balance with the background flow. There is no competition. Perturbations to eddies evolve also in the agreement with the background.

242. Here, the concept of eddy heat flux has to be defined. If you are working with a separate eddy, it may carry heat only because it corresponds to deviations of isotherms from background. But then, this heat is traveling with the eddy. It is obviously not what you are analyzing.

285-286 The sentence implies that the larger velocities are along the minor axis, i.e. zonal. Please edit.

302 vertically

303 The concept of effective depth is questionable, for it would be too naive to expect that the eddies have the same effective depth everywhere.

411 - 414 Nobody had ever thought that eddies are axisymmetric, and showing this was not needed. The manuscript is of interest because it presents the degree of asymmetry.

424-426 (i) The first part is correct, the second is not. (ii) The authors mix isotropic turbulence with anisotropy of vortices. Even in homogeneous isotropic 2D turbulence only the strongest vortices are axisymmetric, but there are multiple elongated filaments. (iii) As I mentioned, there are more sophisticated eddy parameterizations that try to derive diffusivities based on some physics.

426 Who assumed that transports by eddies are isotropic? Of course they depend on background gradients.

431-432 It does not on the present stage. The authors should at least hint how.

485 and 487 -- you repeat the definition twice

489 What do you mean? And why do you use velocities? They do not characterize stratification, you need vertical shear for that.

I do not see good physics in this attempt, especially as concerns the number of eddies and the use of

planetary PV. At least the use of quasigeostrophic PV would be a much better idea.

502 Here the question is to what an extent the correlations between the data (velocity and temperature) which are retrieved from different data sets preserve true correlations. Some critical discussion and explanation is needed. Then, some discussion of effective depth is needed.

Reviewer #2 (Remarks to the Author):

This manuscript is a fairly thorough assessment of the degree of asymmetry in mesoscale eddies, its geographical distribution, and potential consequences for meridional heat transport. The impact of eddy geometry on the transport of physical quantities (heat, salt/freshwater, nutrients, etc.) is a subject that I think has received too little attention. I am glad to see that this work has illustrated some of the connections between planetary/topographic beta and eddy geometry for example; this is information that could be used in parameterizations of eddy transport. There are some minor (but important) clarifications that I would request the authors implement in their revision of the paper. If these changes are made, I am happy to support this paper for publication in Nature Communications.

Clarifications:

Lines 63-66: The along-isopycnal downgradient diffusion the authors are describing is actually the parameterization by Redi (1982). Gent and McWilliams (1990) introduced another parameterization that attempted to capture the diapycnal effect of eddy lateral mixing by reducing the slope of isopycnals (and therefore releasing the potential energy present in the isopycnal slopes). Most ocean models today include both parameterizations. Please correct this description accordingly.

Lines 138-140: How are the major and minor axes determined? The Method section later (lines 469-474) states that the AVISO Mesoscale Eddy Trajectory Atlas is used to identify the eddies, and that dataset includes eddy edge contours. However, this still leaves some questions about how the authors use eddy contours to obtain the major and minor axis of each eddy. Are the major/minor axes defined by looking for the axis orientations associated with a maximum or minimum distance between contours? Since eddies are not perfect ellipses, the major and minor axes identified this way may not be perpendicular, though in the example eddy (Fig. 1b-c) these axes are perpendicular. Or is the major axis identified as the axis with maximum distance across the eddy, and the minor axis is just assumed to be perpendicular?

Line 201: Where does the theoretical expectation about the global distribution of I_d come from? Is there a reference to support this?

Figure 3 caption and lines 484-500: The relative error is lower for the planetary beta only fit, compared to the fit with the other parameters. Presumably this is because of the exponential term in the planetary beta fit, which captures the abrupt change in eddy orientations approaching the equator. What happens if the exponential term for the planetary beta fit is retained in the other fit? Also, what is the relative error reduction associated with each of the terms in the 5-parameter (x_0 - x_4) fit? It would be helpful to know how important each of these terms is in explaining the eddy orientations.

Fig. 4i-j: I was a little confused when I first saw these figures that the mean SST anomaly in a composite anticyclonic eddy seemed to be zero, until I read later that the symmetric component of SST had been removed prior to computing the composites (lines 530-533). It would be helpful for the figure caption to specify that these are the asymmetric SST anomalies.

Line 474: This is a very large number of eddies, even for a 29-year dataset! Please clarify: there are probably not ~ 70 million distinct eddies, but rather 70 million instances of eddy identification from a map of SSH/SLA. With $\sim 10,000$ daily SLA maps this would be about 7000 eddies per map, or about one eddy per 2.5° square of ocean. This seems plausible, but if there are truly 70 million eddies in the

dataset most of them must be very short-lived, and the authors may want to consider filtering for eddies that have a longer duration (e.g., >1 week or >2 weeks).

Line 515: What influenced the choice of $H = 200$ meters? Is there a reference that supports this? We might expect the appropriate H to vary with latitude and stratification, for example. I realize that this is an approximation and the key conclusions of the paper do not depend on the choice of H , so perhaps a brief literature search and a reference or two would be enough to support this choice.

Andrew Delman
University of California Los Angeles

Reviewer #3 (Remarks to the Author):

This manuscript presents the asymmetry feature of mesoscale eddies and the induced heat flux based on satellite data. Overall, this manuscript is well-written and organized. However, additional results and discussions are necessary, and the limitations of this study need to be pointed out. Therefore, I recommend that several major points should be addressed before the manuscript is accepted for publication.

Reviewer #1 (Remarks to the Author):

By processing the available data based on satellite altimetry (geostrophic velocities and sea surface height anomalies) and the data on sea surface temperature the manuscript analyses the deviations of the ocean mesoscale eddies from the circular shape and shows that the elongated form of eddies is essential in defining the eddy induced mean eddy heat flux.

Although these results are of interest, I cannot recommend the manuscript in present form as some of its statements are misleading and require major revision. The authors build their motivation on the argument that the existing eddy parameterizations rely on the assumption that eddies are of circular shape, which is not the case. They mix the notion of axisymmetric eddy with that of isotropic turbulence. The isotropy in the latter case is understood in statistical sense and does not exclude eddies of arbitrary shape. Numerous eddy simulations show the presence of elongated structures in the vorticity field even in isotropic quasi-two dimensional turbulence. Furthermore, asymmetry of eddies translates in effective eddy mixing (which incorporates the effect of multiple random eddies) in a very indirect way. Of course, in many cases modelers use isotropic diffusivities in downgradient parameterizations. However, this is done for simplicity reasons.

There is understanding than mixing is not isotropic, and, for example, Smith and Gent (J.Phys. Oceanogr. 34, 2541--2564, 2004) discuss a mathematical framework to use anisotropic diffusivities.

The authors see their finding in the demonstration that the mesoscale eddies in the ocean do not have an axisymmetric form. However, there were no claims in the literature that they are. Ocean mesoscale eddies are generated through baroclinic and barotropic instabilities which occur in the presence of gradients of potential vorticity. This immediately makes the direction along the gradient and cross the gradient non-equivalent.

The partition of eddy velocity field into axisymmetric and the rest illustrate that the deviations from symmetry are not negligible. However, the value of the exercise with eddy heat flux is not immediately clear to me (and it determines the title of the manuscript!). If one takes only a part of real velocity one gets a smaller heat flux. The doubling mentioned in the title of the manuscript is only the consequence of taking

ARTIFICIALLY a part of perturbation. Heat flux is determined by correlations. The shape of velocity and temperature perturbations is the result of evolving instabilities which release available potential energy and hence are accompanied by heat flux. Why it is of interest to decompose real eddies into symmetric and antisymmetric parts? The symmetric part taken separately is not a solution of dynamical equations. One needs the entire perturbations. There is no use in comparing with non-existing solutions.

In summary, the information presented in the manuscript might be of interest, but motivation of the manuscript is questionable.

We are pleased that the reviewer found our results of interest and we want to thank the reviewer for bringing up important issues to help us improve our manuscript. The arguments that existing eddy parameterizations rely on the assumption of circular-shaped eddies or isotropic turbulence have been removed from the revised manuscript and the related statements have been modified accordingly in Introduction.

The motivation of this paper is to bring attention of the community to the asymmetry of mesoscale eddies and its potential influences. In many past studies based on observational data, eddies have long been treated as circular for simplicity (Roemmich and Gilson, 2001; Chelton et al., 2011a and 2011b; Dong et al., 2014; Gaube et al., 2015 and among others). The asymmetry of the eddy dynamical field (e.g. pressure and flow field) is not fully appreciated, and we are glad that the reviewer agrees that the eddy asymmetry can have important influence on the eddy-induced heat flux. We strived in this study to quantify the degree of eddy asymmetry and its influence on the magnitude of eddy-induced heat flux. The contribution of the asymmetric flow field to the eddy-induced heat flux is found to have almost the same magnitude as their symmetric counterpart. This is not what we expected and is something we want to bring the community's attention to.

Our results also indicate that the elongation of mesoscale eddies is directionally-dependent. On average, there are preferential directions of eddy elongation as shown in the following global average distribution of I_d (the index of the preference direction). The distribution of I_d is influenced by the planetary beta effect, the background flow field and topographic features, which lead to an anisotropic eddy-induced heat flux. The anisotropic viscosity and diffusion have been justified both by laboratory and numerical experiments, which are also influenced by planetary beta

effect, the background flow field and topographic features by Holloway and Kristmannson (1984), Haidvogel and Keffer (1984), Bartello and Holloway (1991), Figueroa and Olson (1994), Smith and McWilliams (2003), and Smith and Gent (2004). Thus, we also hope our results could provide useful observational constrain for future parameterization of eddy-induced heat flux in numerical models.

Figure.R1. Global distribution of eddy direction index $I_a = \sin(\theta)$, where θ is the direction angle of major axis of eddy elongation. This map is constructed by averaging with a 1° moving window at each grid point for all available data points.

Partitioning is a useful method in fluid dynamics, such as the Helmholtz velocity decomposition, which is a partitioning of velocity into its rotational and divergent components. We decompose the eddy velocity field into symmetric and asymmetric parts aiming to highlight the contribution of asymmetric velocities to the eddy-induced heat flux. In fact, the total meridional eddy-induced heat flux (EHF) increases with the asymmetry of mesoscale eddies, as shown in Fig.R2a. At the same time, the meridional heat flux induced by the asymmetric/symmetric parts of the eddy velocity field increases/decreases significantly with the increasing I_a , as shown in Figs.R2c and R2d. The contribution of the asymmetric part is nearly doubled (from 37% to 61%) with increasing I_a from 1 to 3. These results indicate that elongated eddies will induce larger meridional heat flux due to the increasing contribution of its asymmetric part of the velocity field.

Figure.R2. (a) Curve of the total meridional eddy-induced heat fluxes (EHF) changing with the eddy asymmetric index I_a . (b) The proportions of asymmetric and symmetric parts of the total EHF changing with I_a . The red area represents the asymmetric part and the blue area for the symmetric part. The percentage numbers on the left correspond to $I_a = 1$ (axis-symmetric eddy), while the percentage numbers on the right correspond to $I_a = 3$ (extremely elongated eddy). (c) Curve of the meridional EHF induced by asymmetric velocity field changing with I_a . (d) same as (c) but for symmetric part. The gray range represents the error bar in (a)(b)(c), which is calculated as the standard error. The asymmetric index $I_a = a/b$ is introduced as the ratio between the major and minor axis of eddies

The global distribution of the direction index I_d in Fig.R1 shows that: eddies in the low-latitude regions have a preference of zonal elongation, which can hinder the meridional heat flux. In contrast, eddies in the mid- and high-latitude regions have a preference of meridional elongation, which can facilitate the meridional heat flux. In order to quantify eddy asymmetry's influence on meridional EHF in different parts of the world ocean, eddies are divided into two groups: asymmetric eddies with I_a greater than 1.55, and nearly-symmetric eddies with I_a less than 1.55 (the average value of I_a is 1.55). As shown by the ratio of EHF_y in Fig.R3, asymmetric eddies tend to hinder the meridional heat flux in low latitudes where the ratio is smaller than one and the heat flux is almost doubled in the high latitude regions by the eddy

asymmetry. In a globally average sense, the asymmetry increases the eddy-induced meridional heat flux by about 20%. These results highlight the contribution of eddy asymmetry to the meridional heat flux.

Figures R2-R3 and the technical details of their generation have been added to the revised Supplementary Information (Note 2-3 and Figs. S1-S2).

Figure.R3. Global distribution for the ratio of total meridional eddy-induced heat flux EHF_y with I_a greater than 1.55 and less than 1.55.

Minor points:

line 10 or axisymmetric vortex? --- What do you mean?

The statement has been removed from the revised manuscript.

12 are asymmetric and orientative?? --- Do you simply mean that they are elongated?

If yes, just say so.

We have changed “orientative” to “directionally-dependent” in the revised manuscript.

31 redistribution

We thank the reviewer to point this typo and have corrected it.

36 of mass?

We have removed the word “mass” in this sentence.

44-47 But simulations generally show large spread, and I do not think that at present the amount of high-resolution simulations is sufficient to do reliable conclusions.

We agree with the reviewer that when high-resolution simulations resolve the mesoscale eddies, there will be improvements of the simulation of climate variability and meridional heat fluxes (Griffies et al., 2015; Hewitt, et al, 2016; Chassignet et al., 2020). We have revised the statement accordingly in Line 43-50 in the revised manuscript.

53 '...this calls for a urgent need...'? --- Edit

This sentence has been removed.

63 '...density variations cause tilted isopycnals ...' --- What do you mean?

We have revised the statements accordingly in Line 59-64 in the revised manuscript.

65-66 I do not think this is a correct statement: The effect of eddies is parameterized as a product of thickness diffusivity with the slope gradient. The latter takes into account asymmetry. As concerns the diffusivity, it is sometimes taken as a constant, but sometimes is a subject of separate parameterizations. If it is taken as a constant, it is not because we think that eddies are circular, this is just the simplest choice. There are more sophisticated choices that try to work with local balances.

We thank the reviewer to point this out and we have removed this statement from the revised manuscript.

70-71 Nobody has doubts in this. Nobody is really relying on isotropic turbulence

We thank the reviewer to point this out and have removed this sentence.

72 curvature of the Earth is not the same as beta-effect

We agree with the reviewer and have changed this statement in Line 108-110 in the revised manuscript.

73 'spontaneously' --- What do you mean?

We thank the reviewer to point this out and have modified the corresponding statement in Line 69-73 in the revised manuscript.

73-75 Nobody thinks differently, why do you try to prove this?

We thank the reviewer to point this out and have removed this statement accordingly.

77 Which approximation?

We have removed the word “approximation” in the revised manuscript.

78-79 Downgradient diffusion and axisymmetry of vortices are not related generally. Asymmetric vortices may lead to downgradient diffusion, and axisymmetric vortices will not necessarily result in downgradient diffusion.

We thank the reviewer to point this out and we have removed this statement accordingly.

83, 84 Again, this is wrong statement: Eddies always exist of the background and even the onset of instability is related to the structure of PV gradient. It is the presence of background gradients that makes eddies as they are.

We agree with the reviewer that eddies are asymmetric when they emerge from the background PV gradient through instability processes. During the evolution into their mature phase, they will undergo adjustments and axisymmetrization processes (Melander & McWilliams, 1987). We are interested in the degree of asymmetry and its corresponding influence on the heat transports. We have revised the statement accordingly in Line 69-81 in the revised manuscript.

88-90 Taken generally, this is a trivial statement, one would be surprised if it were otherwise. Of course the magnitude of the effect is of interest.

We agree with the reviewer and it is the magnitude we hope to quantify in this study. We have modified the statement in Line 85-87 in the revised manuscript.

92-93 This statement should be corrected: You split velocity into two parts, and there is nothing surprising in what you see. Your symmetric velocity corresponds to smaller energy and smaller heat flux. So what is claimed? That part is smaller than the whole? Eddies are as they are created by dynamics, there is no need in replacing them by symmetric eddies.

The reviewer's understanding of our results is correct. The partitioning, by definition, makes the heat flux induced by the symmetric part smaller than the total heat flux, but the magnitude of contributions from the symmetric and asymmetric parts is not obvious. As we have mentioned above, it is a common practice in past data analysis studies to assume oceanic eddies to be circular. What we try to emphasize in this study is that the asymmetric part's contribution to eddy-induced heat flux can be as large as that of the symmetric part.

102, 103 f-plane is not what is written here

We thank the reviewer to point this out and have modified the statement accordingly in Line 94-95 in the revised manuscript.

105-106 Only in conditions of weak background gradients.

We thank the reviewer to point this out and have modified the statement in Line 96-99 in the revised manuscript.

106-107 Freely evolving turbulence has very little to do with oceanic turbulence (and please use quasi-geostrophic)

We thank the reviewer to point this out and have modified the statement accordingly in Line 99 in the revised manuscript.

108-109 May be in simple theories, but not otherwise.

We have modified the statement accordingly in Line 100-101 in the revised manuscript.

132 '...rotational effect of stratified fluid is not homogenous due to the Earth's surface curvature' --- please provide a correct definition of PV. Why are you talking about Earth's curvature? Do you mean Earth sphericity? Or simply the latitudinal dependence of the Coriolis parameter?

We thank the reviewer to point this out and have modified the statement accordingly in Line 109-111 in the revised manuscript.

166-167 Do you mean planetary PV gradient? In reality PV gradient depends on vertical velocity shear and the gradient of relative vorticity of background flow. Why

everything should depend on planetary PV gradient? It is not necessarily dominant in many places. The Rhines scale is obviously related to flat-bottom barotropic turbulence, and its appearance here is questionable at least.

We thank the reviewer to bring up this point. By PV gradient, we mean the PV gradient caused by all dynamical processes. The corresponding statements have been modified in Line 129-132 in the revised manuscript.

The bootstrap test on fitting of the index I_d (see Supplementary Information Note 1 and Table S1 for details) suggests that the planetary beta effect plays a dominant role in regulating the eddy elongation, and the Rhine scale is introduced as an index to represent the regional relative importance of the planetary beta effect.

182-183 This is a recurrent phrase, but the ability of an eddy to preserve its shape depends on background state, as I have already mentioned.

We thank the reviewer to bring up this point and have modified the statement accordingly in Line 144-145 in the revised manuscript.

191 "The elongated eddies have an orientative nature"? Are you willing to say that elongated eddies can be characterized by the direction of major axis to zonal direction? If yes, then say

The reviewer is correct and we have modified the statement in Line 153-154 in the revised manuscript.

193 has

We have corrected the typo.

210 PV gradient is not only beta-effect.

We thank the reviewer to point this out and we modified the statement in Line 168-171 in the revised manuscript.

224 'with high density of crowded eddies' -- with high density of eddies

226-227 But why? If there are many eddies, the strain created by them will be randomly distributed.

The eddy number has been removed from the factors used to fit the index I_d , and the corresponding statements in the revised manuscript have also been modified.

234-236 It is a repetitive statement which contains no information without writing formulas. Eddies are always in balance with the background flow. There is no competition. Perturbations to eddies evolve also in the agreement with the background.

This sentence has been removed from the revised manuscript.

242. Here, the concept of eddy heat flux has to be defined. If you are working with a separate eddy, it may carry heat only because it corresponds to deviations of isotherms from background. But then, this heat is traveling with the eddy. It is obviously not what you are analyzing.

The eddy induced heat flux Q is defined as:

$$Q = \overline{V'T'} = (\overline{u'T'}, \overline{v'T'}) \quad (R1)$$

where u' and v' are the zonal and meridional geostrophic current anomalies given by altimeter data; T' is the temperature anomaly given by the instantaneous satellite observations of sea surface temperature minus the climatological monthly average temperature from 1993 to 2019. The overbar in (R1) denotes average within a square with the size ± 2 eddy radii around the eddy center. The detailed information can be found in the Method section.

The eddy heat transport can be decomposed into trapping and stirring components by partitioning of the temperature anomaly field into symmetric and asymmetric parts (Frenger et al., 2015; Abernathey and Haller, 2018). Since the main focus of our work is on asymmetry of eddy's dynamical structure (velocity and pressure field) and its influence on the heat transports, we did not separate the temperature anomaly field. Thus, the Q defined in (R1) denotes the total eddy induced surface heat flux, containing both the stirring and trapping effects.

285-286 The sentence implies that the larger velocities are along the minor axis, i.e. zonal. Please edit.

We thank the reviewer to point this out and have correct the statement in Line 204-206 in the revised manuscript.

302 vertically

The sentence is removed from the revised manuscript.

303 The concept of effective depth is questionable, for it would be too naive to expect that the eddies have the same effective depth everywhere.

We agree with the reviewer that the effective depth is a very rough approximation, and it should vary globally. Considering the vertically integrated heat flux is not the main point of this paper, we no longer use the effective depth and try to compute the vertically integrated heat flux in the revised manuscript. Instead, we have focused in the revised manuscript on the surface properties only. The corresponding statements and figures have been modified in the revised manuscript.

411 - 414 Nobody had ever thought that eddies are axisymmetric, and showing this was not needed. The manuscript is of interest because it presents the degree of asymmetry.

We thank the reviewer to point this out and have modified the statement in Line 281-284 in the revised manuscript.

424-426 (i) The first part is correct, the second is not. (ii) The authors mix isotropic turbulence with anisotropy of vortices. Even in homogeneous isotropic 2D turbulence only the strongest vortices are axisymmetric, but there are multiple elongated filaments. (iii) As I mentioned, there are more sophisticated eddy parameterizations that try to derive diffusivities based on some physics.

426 Who assumed that transports by eddies are isotropic? Of course they depend on background gradients.

We thank the reviewer to point this out and have removed the corresponding statements from the revised manuscript.

431-432 It does not on the present stage. The authors should at least hint how.

We thank the reviewer to bring this up and have modified the statement accordingly in Line 297-304 in the revised manuscript.

485 and 487 -- you repeat the definition twice

We thank the reviewer to point this out and we have modified the statement in Line 360-362 in the revised manuscript.

489 What do you mean? And why do you use velocities? They do not characterize stratification, you need vertical shear for that. I do not see good physics in this attempt, especially as concerns the number of eddies and the use of planetary PV. At least the use of quasi-geostrophic PV would be a much better idea.

Geostrophic velocity is commonly small at deep depth compared to its surface value. In the widely-used reduced gravity model, surface geostrophic velocity $(\overline{u_g}, \overline{v_g})$ can approximately represent the vertical shear of geostrophic velocity and it is related to the horizontal gradient of density through the thermal wind relation. Thus, the gradient of PV caused by stratification can be represented approximately by the surface geostrophic current velocity (McDowell et al., 1982; Keffer, 1985; Vallis, 2006).

502 Here the question is to what an extent the correlations between the data (velocity and temperature) which are retrieved from different data sets preserve true correlations. Some critical discussion and explanation are needed. Then, some discussion of effective depth is needed.

We agree with the reviewer that the correlation between velocity and temperature need independent observational data to confirm. The results in our main text are obtained from satellite remote sensing data. In order to rule out the potential bias caused by the geostrophic assumption and the remote-sensing sea surface temperature, we use the surface drifter data to double check our results about the eddy-induced heat flux, which can observe concurrently the absolute surface velocity (at about 15-m average) and the sea surface temperature at depth about 30 cm. The eddy heat flux results by using the surface drifter data are consistent with the results by using the satellite remote sensing data. Please find the detailed comparison in the Supplementary Information Fig.S9.

As explained above, we no longer use the effective depth in our revised manuscript.

Reviewer #2 (Remarks to the Author):

This manuscript is a fairly thorough assessment of the degree of asymmetry in mesoscale eddies, its geographical distribution, and potential consequences for meridional heat transport. The impact of eddy geometry on the transport of physical quantities (heat, salt/freshwater, nutrients, etc.) is a subject that I think has received too little attention. I am glad to see that this work has illustrated some of the connections between planetary/topographic beta and eddy geometry for example; this is information that could be used in parameterizations of eddy transport. There are some minor (but important) clarifications that I would request the authors implement in their revision of the paper. If these changes are made, I am happy to support this paper for publication in *Nature Communications*.

We thank the reviewer for the positive assessment regarding the significance and content of our study. We are particularly pleased that the reviewer feels that our study would be useful for future parameterizations of eddy transport and support our paper for publication in *Nature Communications*. We also want to thank the reviewer for bringing up several important points to help us improve our manuscript.

Clarifications:

Lines 63-66: The along-isopycnal downgradient diffusion the authors are describing is actually the parameterization by Redi (1982). Gent and McWilliams (1990) introduced another parameterization that attempted to capture the diapycnal effect of eddy lateral mixing by reducing the slope of isopycnals (and therefore releasing the potential energy present in the isopycnal slopes). Most ocean models today include both parameterizations. Please correct this description accordingly.

We thank the reviewer to point this out. We have modified the statements following reviewer's comments in Line 59-60 in the revised manuscript.

Lines 138-140: How are the major and minor axes determined? The Method section later (lines 469-474) states that the AVISO Mesoscale Eddy Trajectory Atlas is used to identify the eddies, and that dataset includes eddy edge contours. However, this still leaves some questions about how the authors use eddy contours to obtain the major and minor axis of each eddy. Are the major/minor axes defined by looking for the axis orientations associated with a maximum or minimum distance between contours? Since eddies are not perfect ellipses, the major and minor axes identified this way

may not be perpendicular, though in the example eddy (Fig. 1b-c) these axes are perpendicular. Or is the major axis identified as the axis with maximum distance across the eddy, and the minor axis is just assumed to be perpendicular?

We thank the reviewer to bring up this point and we have added definition for the major and minor axes in Line 349-357 in the Method section.

Line 201: Where does the theoretical expectation about the global distribution of I_d come from? Is there a reference to support this?

We thank the reviewer to point this out and have removed the sentence “which is identical to the theoretical expectations” in Line 161-162 in the revised manuscript.

Figure 3 caption and lines 484-500: The relative error is lower for the planetary beta only fit, compared to the fit with the other parameters. Presumably this is because of the exponential term in the planetary beta fit, which captures the abrupt change in eddy orientations approaching the equator. What happens if the exponential term for the planetary beta fit is retained in the other fit? Also, what is the relative error reduction associated with each of the terms in the 5-parameter (x_0 - x_4) fit? It would be helpful to know how important each of these terms is in explaining the eddy orientations.

Following the suggestions by the reviewer, we have tested the exponential fitting for the other terms, and the results did not improve significantly. In our original fitting, the zonal average orientation index I_d could be well fitted by the planetary beta with the fitted expression:

$$\bar{I}_d = f(\beta) = 0.7598 - 0.1142\beta - 1.3558e^{19.0726\beta - 20.7233}$$

where $\beta = \cos(\text{latitude})$. We have tried to use similar exponential term to fit other factors. The exponential expression is:

$$y = a + bx + ce^{dx}$$

where y is the global orientation distribution I_d and x represents the terms used in linear fitting before, including β_b , β_g and eddy number. After that, we used all the factor fittings with exponential terms in a linear fitting to construct the final global eddy orientation, as shown in Fig.R4. The fitting correlation coefficient is 0.73 and

the relative error is 35%, which does not improve significantly compared with the original fitting.

Figure.R4. (a) The reconstructed global distribution of I_d . (b) The scatter compared the observed and reconstructed eddy orientation index I_d . The correlation coefficient is 0.73 and the relative error is 35%.

Table.R1. The first row shows the factors used to reconstruct the global distribution of orientation index. The corresponding columns are the bootstrap test results, using correlation coefficient (Corr. Coeff.) and relative error (Rel. Err.) to test whether the factors have significant contributions to the reconstruction. If the correlation coefficient/relative error of the tested term is larger/smaller than the 95% of the random group outputs, contribution of the corresponding factor is significant.

Fitting Effect	β_{bx}	β_{by}	β_{gx}	β_{gy}	N_e	$f(\beta)$
Random 95% Corr. Coeff.	0.7240	0.7260	0.7199	0.7218	0.7313	0.6539
Tested Term Corr. Coeff.	0.7381	0.7381	0.7381	0.7381	0.7381	0.7381
Random 95% Rel. Err.	36.36%	35.12%	35.69%	35.39%	36.33%	42.21%
Tested Term Rel. Err.	35.50%	35.50%	35.50%	35.50%	35.50%	35.50%

A bootstrap test is conducted on the fitting factors used to reconstruct the index I_d to determine whether the contributions of each factor are significant. For each factor, its global distribution is disordered randomly, at the same time, keeping the other variables unchanged. The I_d is then reconstructed and the correlations and relative

errors are re-evaluated. This process is repeated for 10,000 times to generate the random compare group. If the correlation coefficients and errors are better than the 95% randomly generated correlation coefficients and errors, the contribution of the corresponding factor is taken as significant. As shown in Table.R1, all variables pass the 95% significance test. Meridional topography gradient (β_{by}) and meridional surface geostrophic current (β_{gy}) don't pass the relative error test. The topography mainly influences eddy orientation in the continental boundaries and mid-ocean ridges, where the topography gradient is mainly zonally orientated in these regions. Thus, β_{bx} is more significant than β_{by} . Zonal geostrophic current is generally stronger than the meridional current, resulting in the weak effect of β_{gy} on the eddy orientation. The eddy number makes relative small contribution to the reconstruction.

Fig. 4i-j: I was a little confused when I first saw these figures that the mean SST anomaly in a composite anticyclonic eddy seemed to be zero, until I read later that the symmetric component of SST had been removed prior to computing the composites (lines 530-533). It would be helpful for the figure caption to specify that these are the asymmetric SST anomalies.

We thank the reviewer to point this out and we have added the corresponding statements into the revised figure caption.

Line 474: This is a very large number of eddies, even for a 29-year dataset! Please clarify: there are probably not ~70 million distinct eddies, but rather 70 million instances of eddy identification from a map of SSH/SLA. With ~10,000 daily SLA maps this would be about 7000 eddies per map, or about one eddy per 2.5° square of ocean. This seems plausible, but if there are truly 70 million eddies in the dataset most of them must be very short-lived, and the authors may want to consider filtering for eddies that have a longer duration (e.g., >1 week or >2 weeks).

We thank the reviewer to point this out and the eddy dataset used here only provides the identification of eddies with life-span longer than 10 days. Thus, we have modified the corresponding statements in Line 336-340 in the revised manuscript.

Line 515: What influenced the choice of $H = 200$ meters? Is there a reference that supports this? We might expect the appropriate H to vary with latitude and stratification, for example. I realize that this is an approximation and the key conclusions of the paper do not depend on the choice of H , so perhaps a brief literature search and a reference or two would be enough to support this choice.

We agree with the reviewer that the effective depth is a very rough approximation, and it should vary globally. Considering the vertically integrated heat flux is not the main point of this paper, we no longer use the effective depth and try to compute the vertically integrated heat flux in the revised manuscript. Instead, we have focused in the revised manuscript on the surface properties only. The corresponding statements and figures have been modified in the revised manuscript.

Reviewer #3 (Remarks to the Author):

This manuscript presents the asymmetry feature of mesoscale eddies and the induced heat flux based on satellite data. Overall, this manuscript is well-written and organized. However, additional results and discussions are necessary, and the limitations of this study need to be pointed out. Therefore, I recommend that several major points should be addressed before the manuscript is accepted for publication.

We thank the reviewer for the positive assessment of our study and for bringing up several important points to help us improve our manuscript.

1 The global pattern of the observed and reconstructed eddy orientation index is compared in Fig. 3. The authors have only considered the factor of eddy number to reflect the effect of eddy-eddy interaction. Actually, using the relative strength of the target eddy and the background eddy may be a better indicator. For instance, when a strong eddy encounters a weak eddy, the weak eddy may be more prone to deformation and exhibit asymmetric features. In addition, I am wondering if the asymmetry feature varies in different life stages of mesoscale eddies (e.g., growing, stable, and decaying).

We thank the reviewer to bring this point up and, following the reviewer's suggestion, we have tested the relative strength of target eddy and the background eddy as an indicator of eddy-eddy interaction. The relative interaction strength is calculated as: First, for a target eddy, eddies within its four times radii are identified and their amplitude values are added; Then divide the added amplitude by the amplitude of the target eddy. In this way, there is a ratio In_{ee} for each eddy to evaluate the relative interaction strength. By creating a latitude and longitude grid with $1^\circ \times 1^\circ$ resolution and a moving average window size 1° , the ratio In_{ee} is shown in Fig.R5c. Using this eddy-eddy interaction map to fit the global eddy orientation distribution and the results are shown in Fig.R5. The fitting relative error is 37% and the correlation coefficient is 0.73. There is no significant improvement compared with the fitting that simply used eddy number.

Figure.R5. (a) The observed global distribution of eddy orientation index I_d . (b) The reconstructed global distribution of I_d . (c) The global distribution of eddy-eddy interaction intensity In_{ee} , which is used to reconstruct global distribution of I_d . (d) The scatter compared the observed and reconstructed eddy orientation index I_d . The correlation coefficient is 0.73 and the relative error is 37%.

A bootstrap test is conducted on the fitting factors used to reconstruct the orientation index I_d to determine whether the contributions of each factor are significant. For each factor, its global distribution data is disordered randomly, at the same time, keeping the other variables unchanged. The I_d is then reconstructed and the correlation and relative errors are re-evaluated. This process is repeated for 10,000 times to generate the random compare group. If the correlation coefficients and errors are better than the 95% randomly generated correlation coefficients and errors, the contribution of the corresponding factor is taken as significant. As shown in Table.R2., all variables pass the correlation coefficient test. But β_{by} , β_{gy} and In_{ee} do not pass the relative error test.

Considering that the eddy-eddy interaction factor (e.g. eddy number or relative strength) does not contribute significantly to the fitting, we no longer use this factor to do the fit of I_d in the revised manuscript.

Table.R2. The first row shows the factors used to reconstruct the global distribution of orientation index. The corresponding columns are the bootstrap test results, using correlation coefficient (Corr. Coeff.) and relative error (Rel. Err.) to test whether the factors have significant contributions to the reconstruction. If the correlation coefficient/relative error of the tested term is larger/smaller than the 95% of the random group outputs, contribution of the corresponding factor is significant.

Fitting Effect	β_{bx}	β_{by}	β_{gx}	β_{gy}	I_{nee}	$f(\beta)$
Random 95% Corr. Coeff.	0.7152	0.7183	0.7070	0.7137	0.7295	0.4591
Tested Term Corr. Coeff.	0.7313	0.7313	0.7313	0.7313	0.7313	0.7313
Random 95% Rel. Err.	37.75%	36.59%	38.37%	36.87%	36.41%	52.23%
Tested Term Rel. Err.	36.95%	36.95%	36.95%	36.95%	36.95%	36.95%

Figure.R6. The average I_a change during eddy's lifespan. The red solid line represents anticyclonic eddy and the blue dashed line represents cyclonic eddy. The x-axis is the normalized lifespan, which interpolate different length of eddy life into the same length.

We also want to thank the reviewer for the insightful suggestion about asymmetry feature varying in different life stages of mesoscale eddies. Our additional analyses indicate that eddy asymmetry could change during its lifespan. Different eddy has different length of lifespan. Thus, the I_a of eddies with different length of lifespan is normalized into the same length of lifespan from 0 to 1 through interpolation. Then take the average of all the I_a change. The averaged curves of I_a

is then composited against the normalized time by eddy lifespan as shown in Fig.R6. During the growing and decaying stage, eddies have the strongest asymmetry, corresponding to a relatively large I_a . When eddies become mature, they have a smaller I_a and high symmetry. We have added this result into the revised Supplementary Information (Note 4 and Fig.S3).

2 Hausmann and Czaja (2012) previously estimated the mixed-layer heat transport by mesoscale eddies using SST and SSH observations, and they already presented the asymmetric pattern of meridional heat transport (their Fig. 6) due to eddy swirl. Although the authors cite this work, they do not explicitly mention the key results in the introduction, which is quite related to the present study. Moreover, the reference for the calculation formula of eddy flux (Eq. 7) should be clearly indicated.

3 The eddy heat transport is decomposed into symmetric and asymmetric components. From my perspective, these components correspond exactly to eddy trapping and eddy stirring, respectively, as illustrated by Frenger et al. (2015) in their Fig. 10. In addition, a recent study by Abernathey and Haller (2018) found that the eddy stirring makes a dominant contribution to the meridional eddy flux, which is far greater than “doubling” estimated by this study. Hence, it is essential to provide clear physical interpretations of both components and discuss the relationship with previous studies.

We thank the reviewer to bring this point up. The former studies about the asymmetry of mesoscale eddies are mainly about the sea surface temperature anomaly (SSTA) field. Hausmann and Czaja (2012) tried to quantify the “swirl” and “drift” heat fluxes induced by eddies by investigating the asymmetry of the composite SSTA field of mesoscale eddies, as shown in their Figs.4-5. From their figures, the nearly axis-symmetric SSHA field can be identified. Frenger et al. (2015) decomposed the eddy-induced heat transport into trapping and stirring components by partitioning the SSTA field into symmetric (Monopole) and asymmetric (Residual) parts, as shown in their Figs.9-10. From their figures, the nearly axis-symmetric SSHA field can be identified again. We also agree that the stirring components can be much larger than the trapping components as suggested by Abernathey and Haller (2018).

Because the main focus of our work is on the asymmetry of eddy’s dynamical structure (velocity and pressure field) and its influence on the heat transports, we preferred not to separate the temperature anomaly field. Thus, the eddy induced heat

flux Q defined in our paper is the total eddy induced surface heat flux, which contains both the stirring and the trapping effect. When we separate the eddy velocity field into symmetric and asymmetric field, they do not necessarily correspond exactly to eddy trapping and eddy stirring. Considering the axis-symmetric velocity field and a monopole SSTA field, the resultant heat flux turns to be zero, which does not reflect the eddy trapping effect. Additional statements and discussions have also been added to the revised manuscript in Line 77-81 and Line 272-278.

4 The calculation of the vertical integrated eddy-induced heat flux is based on an effective depth ($H = 200$ m). It would be helpful to explain the physical basis for choosing this depth. Is it an arbitrary choice, or is there a physical basis behind it? Moreover, it is crucial to investigate whether the adoption of a realistic effective depth will lead to consistent heat flux estimates in the Southern Ocean with those reported in previous studies (Lines 335-337). The primary focus of this paper is on the SURFACE (or mixed-layer) asymmetry of mesoscale eddies using satellite observations. If there is insufficient evidence to demonstrate the validity of these results in the subsurface, this limitation should be explicitly stated in the main conclusion and title. Otherwise, the authors' claims may be exaggerated.

We thank the reviewer to bring this point up. The effective depth was introduced by Qiu and Chen (2005) in the subtropical countercurrent region as 200 m based on subsurface observations. We agree with the reviewer that the effective depth should vary globally and may cause a bias on the magnitude of the vertical integrated eddy-induced heat flux. Considering the vertically integrated heat flux is not the main point of this paper, we no longer use the effective depth and try to compute the vertically integrated heat flux in the revised manuscript. Instead, we have focused in the revised manuscript on the surface properties only. The corresponding statements and figures have been modified in the revised manuscript.

We have also changed the title to “Doubling of Surface Oceanic Meridional Heat Transport by Non-Symmetry of Mesoscale Eddies”.

5 Several specific points.

(1) Figures. Explain the white area near the equator.

We thank the reviewer to bring this point up. Mesoscale eddies are basically quasi-geostrophic motions. When the Coriolis parameter is too small near the equator, the

geostrophic balance does not hold and the geostrophic flows can no longer be estimated.

(2) Line 244. each category “occupies”.

We have corrected the typo.

(3) Line 506. It is not clear how to obtain T' . Instantaneous SST minus the monthly mean?

We thank the reviewer to point this out and have modified the statement in Line 378-380 in the revised manuscript to clarify how T' is calculated.

Reference

1. Bartello, P. & Holloway, G. Passive scalar transport in beta-plane turbulence. *J. Fluid Mech.*, **223**, 521–536 (1991).
2. Chassignet, E. P. et al. Impact of horizontal resolution on global ocean–sea ice model simulations based on the experimental protocols of the Ocean Model Intercomparison Project phase 2 (OMIP-2). *Geosci. Model. Dev.* **13**, 4595-4637 (2020).
3. Chelton, D. B., Gaube, P., Schlax, M. G., Early, J. J., & Samelson, R. M. The influence of nonlinear mesoscale eddies on near-surface oceanic chlorophyll. *Science* **334(6054)**, 328-332 (2011a).
4. Chelton, D. B., Schlax, M. G., & Samelson, R. M. Global observations of nonlinear mesoscale eddies. *Prog. Oceanogr.* **91(2)**, 167-216 (2011b).
5. Dong, C., McWilliams, J. C., Liu, Y., & Chen, D. Global heat and salt transports by eddy movement. *Nat. Commun.* **5(1)**, 3294 (2014).
6. Figueroa, H. A. & Olson, D. B. Eddy resolution versus eddy diffusion in a double gyre GCM. Part I: The Lagrangian and Eulerian description. *J. Phys. Oceanogr.*, **24**, 387–401 (1994).
7. Gaube, P., Chelton, D. B., Samelson, R. M., Schlax, M. G. & O’Neill, L. W. Satellite observations of mesoscale eddy-induced Ekman pumping. *J. Phys.*

- Oceanogr.*, **45**, 104-132 (2015).
8. Gent, P. R., & McWilliams, J. C. Isopycnal mixing in ocean circulation models. *J. Phys. Oceanogr.* **20**(1), 150-155 (1990).
 9. Griffies, S. M. et al. Impacts on Ocean heat from Transient Mesoscale Eddies in a Hierarchy of Climate Models. *J. Climate*. **28**, 952-977.
 10. Haidvogel, D. B. & Keffer, T. Tracer dispersal by mid-ocean mesoscale eddies. *Dyn. Atmos. Oceans*, **8**, 1–40 (1984).
 11. Hewitt, H. T., et al. Resolving and parameterising the ocean mesoscale in Earth system models. *Curr. Clim. Change Rep.* **6**, 137–152 (2020).
 12. Holloway, G., & Kristmannsson, S. Stirring and transport of tracer fields by geostrophic turbulence. *J. Fluid Mech.*, **141**, 27–50 (1984).
 13. Keffer, T. The ventilation of the world's oceans: Maps of the potential vorticity field. *J. Phys. Oceanogr.* **15**, 509-523(1985).
 14. McDowell, S., Rhines, P., & Keffer, T. North Atlantic potential vorticity and its relation to the general circulation. *J. Phys. Oceanogr.* **12**, 1417-1436 (1982).
 15. Melander, M. V., McWilliams, J. C., & Zabusky, N. J. Axisymmetrization and vorticity-gradient intensification of an isolated two-dimensional vortex through filamentation. *J. Fluid Mech.* **178**, 137-159 (1987).
 16. Qiu, B., & Chen, S. Eddy-induced heat transport in the subtropical North Pacific from Argo, TMI, and altimetry measurements. *J. Phys. Oceanogr.* **35**(4), 458-473 (2005).
 17. Redi, M. H. Oceanic Isopycnal Mixing by Coordinate Rotation. *J. Phys. Oceanogr.* **12**, 1154-1158 (1982).
 18. Roemmich, D., & Gilson, J. Eddy transport of heat and thermocline waters in the North Pacific: A key to interannual/decadal climate variability? *J. Phys. Oceanogr.* **31**(3), 675-687 (2001).
 19. Smith, R.D. & McWilliams, J. C. Anisotropic horizontal viscosity for ocean models. *Ocean Modell.*, **5**, 129–156 (2003).
 20. Smith, R.D. & Gent, P. R. Anisotropic Gent–McWilliams Parameterization for Ocean Models. *J. Phys. Oceanogr.*, **34**, 2541–2564 (2004).

21. Vallis, G. K., 2006: *Atmospheric and Oceanic Fluid Dynamics*, Cambridge University Press, 745 pp.
22. Wiebe, E. C., & Weaver, A. J. On the sensitivity of global warming experiments to the parametrization of sub-grid scale ocean mixing. *Clim. Dyn.* **15**, 875-893 (1999).
23. Zhang, Z., Zhang, Y., Wang, W., & Huang, R. Universal structure of mesoscale eddies in the ocean. *Geophysical Research Letters*, 40:, 3677-3681(2013).
24. Zhang, Z., Wang, W., & Qiu, B. Oceanic mass transport by mesoscale eddies. *Science* **345(6194)**, 322-324 (2014).

REVIEWER COMMENTS

Reviewer #1 (Remarks to the Author):

The revised version has eliminated many misleading places. I, however, still cannot accept the decomposition into symmetric-asymmetric parts in the section "Heat flux influenced by eddy asymmetry". Indeed, the symmetric part the authors try to use is not a solution of equations of motion. It is an artificial object created by the authors. There cannot be symmetric eddies in places with PV gradients unless one deals with a limiting case of very strong eddies. The decomposition the authors are advocating is a misconception, which permeates the manuscript. There are asymmetric eddies, and they are the entities one is dealing in the nature. There is no way to make them symmetric, so it is against any physical wisdom to claim that eddy asymmetry increases the heat flux.

The value of the manuscript is in showing in a systematic way the distribution of eddies and analysis of mean eddy heat flux. This information can be of interest for modellers to tune their eddy-resolving setups. I am not sure that this alone merits the publication in this journal. The remark that eddy fluxes are not necessarily downgradient is also a valid one, but this is well known.

Reviewer #2 (Remarks to the Author):

Thanks to the authors for their response to my comments and associated revisions. I would note that I agree with a key point of the authors' response to Reviewer 1. While it has been common knowledge that eddies are not always symmetric, there have been a number of eddy-composite studies that assume circular eddy shapes, and not much attention given to the effects of anisotropy in eddies. There are many reasons why we might care about asymmetry or anisotropy in eddies, apart from how they are parameterized in coarse-resolution ocean models. I do have recommended changes to one part of the manuscript, but with those changes would recommend it for publication.

Recommended edits:

Lines 359-372 and Figure 3: Some aspects of the reconstruction of the directional index are still puzzling to me, and might be to readers as well. If the relative error of the reconstruction using planetary beta only has a 26% relative error and the reconstruction using all of the parameters has ~35% relative error, what is gained by using the other parameters? Table S.1 shows the results of the bootstrap analysis, which also confirms that planetary beta by far is the most important factor in the reconstructions, though the others are close to the 95% significance thresholds. I have some recommendations for presenting this in a clearer way:

- Show the p-values of significance for each of the individual factors, rather than just the 95% significance thresholds. This can be readily done from a bootstrap distribution of values, and would give a sense of how important the difference is between a 36.25% RE 95% sig threshold and an actual 36.47% RE, for example.
- I presume the authors must have included the extra parameters because they increased the correlation of the reconstruction (relative to using planetary beta only), despite the increase in RE. That would suggest that the extra parameters are improving the fit in some cases while adding unnecessary random "noise" in others. Could the authors show (possibly in a map) where using the additional parameters improves the fit vs. where it does not?

Andrew Delman
University of California Los Angeles

Reviewer #3 (Remarks to the Author):

I am delighted to observe that the author has conscientiously addressed my comments and suggestions in their revised manuscript. Their modifications have considerably enhanced the clarity and coherence of the paper. This work holds substantial significance in advancing our understanding of the structure of mesoscale eddies and the eddy-induced heat transport. With the paper now in good shape, I am glad to recommend its publication in Nature Communications after addressing the following minor points.

(1) L50 and L250. The abbreviation of EKE is redundantly defined.

(2) L333. Indicate the version of the eddy dataset to facilitate reproducibility of your work by readers.

(3) Figures. I understand that the white area means the region does not hold the geostrophic balance. This should be indicated in the caption of the first figure.

Reviewer #1 (Remarks to the Author):

The revised version has eliminated many misleading places. I, however, still cannot accept the decomposition into symmetric-asymmetric parts in the section "Heat flux influenced by eddy asymmetry". Indeed, the symmetric part the authors try to use is not a solution of equations of motion. It is an artificial object created by the authors. There cannot be symmetric eddies in places with PV gradients unless one deals with a limiting case of very strong eddies. The decomposition the authors are advocating is a misconception, which permeates the manuscript. There are asymmetric eddies, and they are the entities one is dealing in the nature. There is no way to make them symmetric, so it is against any physical wisdom to claim that eddy asymmetry increases the heat flux.

The value of the manuscript is in showing in a systematic way the distribution of eddies and analysis of mean eddy heat flux. This information can be of interest for modellers to tune their eddy-resolving setups. I am not sure that this alone merits the publication in this journal. The remark that eddy fluxes are not necessarily downgradient is also a valid one, but this is well known.

We appreciate that the reviewer found valuable points and potential applications of our paper and we want to thank the reviewer for bringing up important issues to help us improve our manuscript.

We understand reviewer's point that when it is weak in a strong PV gradient background, an eddy cannot sustain its axisymmetric shape and will be asymmetrically elongated. Instead of analytic solutions based on fundamental equations, there are now broad existing literatures that employ eddy-tracking algorithms to study the characteristics of mesoscale eddies (Chelton et al., 2007; 2011a; 2011b; Dong et al., 2014; Gaube et al., 2015). Such approaches may not stem directly from the fundamental equations but can be a useful addition to theoretical studies. For example, Chelton et al. (2011a) pointed out that most eddies are strong enough to sustain nonlinear coherent structures. In Fig.2 of our manuscript, the distribution of I_a also confirms that a substantial part of all observed eddies are very close to circular shapes.

We agree with the Reviewer 2's assessment that there have been a number of eddy-composite studies that have assumed circular eddy shapes. It is our hope that our present study will bring attention of the research community that it is important to quantify the eddy anisotropy not only geometrically in its shapes but also quantitatively in its contribution to meridional heat transport.

Reviewer #2 (Remarks to the Author):

Thanks to the authors for their response to my comments and associated revisions. I would note that I agree with a key point of the authors' response to Reviewer 1. While it has been common knowledge that eddies are not always symmetric, there have been a number of eddy-composite studies that assume circular eddy shapes, and not much attention given to the effects of anisotropy in eddies. There are many reasons why we might care about asymmetry or anisotropy in eddies, apart from how they are parameterized in coarse-resolution ocean models.

I do have recommended changes to one part of the manuscript, but with those changes would recommend it for publication.

We thank the reviewer for the positive assessment regarding the significance and content of our study. We are particularly pleased that the reviewer agrees with us that more attention should be given to the effects of anisotropy in eddies and supports our paper for publication in *Nature Communications*.

Recommended edits:

Lines 359-372 and Figure 3: Some aspects of the reconstruction of the directional index are still puzzling to me, and might be to readers as well. If the relative error of the reconstruction using planetary beta only has a 26% relative error and the reconstruction using all of the parameters has ~35% relative error, what is gained by using the other parameters? Table S.1 shows the results of the bootstrap analysis, which also confirms that planetary beta by far is the most important factor in the reconstructions, though the others are close to the 95% significance thresholds. I have some recommendations for presenting this in a clearer way:

- Show the p-values of significance for each of the individual factors, rather than just the 95% significance thresholds. This can be readily done from a bootstrap distribution of values, and would give a sense of how important the difference is between a 36.25% RE 95% sig threshold and an actual 36.47% RE, for example.
- I presume the authors must have included the extra parameters because they increased the correlation of the reconstruction (relative to using planetary beta only), despite the increase in RE. That would suggest that the extra parameters are improving the fit in some cases while adding unnecessary random "noise" in others. Could the authors show (possibly in a map) where using the additional parameters improves the fit vs. where it does not?

We thank the reviewer to bring up this point. If the global distribution of directional index I_d is fitted only by the planetary beta, the relative error would be ~38%, which is larger than the ~35% relative error of reconstruction using all of the parameters. The ~26% relative error is of the reconstruction of zonal-averaged curve of I_d rather than the global distribution map, thus cannot be compared with the all-parameter reconstruction relative error of ~35%. We have removed this confusing statement about the ~26% relative error in the revised manuscript in Line 363. The global map of I_d reconstructed only by planetary beta is shown in Fig.R1a, which exhibits zonally uniform stripped distribution, and misses a lot of detailed fine structures of the I_d global distribution reconstructed by all parameters as shown in Fig.R1b.

Following the suggestions by reviewer, the p -values of bootstrap test for individual factors are computed and shown in Table.R1 below. If the p -value of a coefficient is smaller than 0.05, this term is significant at the 95% significance level. The p -values in Table.R1 are all well below this threshold, indicating that all factors we adopted make significant contributions to the reconstruction. At the same time, the p -values for each term are all too small to determine the relative contributions of each term. Further t -test of the fitting is conducted and the results are also shown in Table.R1. For the t -test, if the t -statistic is larger than 2.57, the result would reach a 95% significant level in a linear fitting with five coefficients. The values in Table.R1 are also well above this threshold, and the planetary beta term has the largest t -statistic value suggesting its most important contribution. The corresponding statements of above have been added to the revised Supplementary Information in Supplementary Note 1 and Table.S1.

Table.R1. The first row shows the factors used to reconstruct the global distribution of directional index. The second to six rows are the bootstrap test results, using correlation coefficient (Corr. Coeff.) and relative error (Rel. Err.) to test whether the factors have significant contributions to the reconstruction. If the correlation coefficient/relative error of the tested term is larger/smaller than the 95% of the random group outputs, contribution of the corresponding factor is significant. The p -value in the six row is computed by one-sided empirical method. The seven row is the results of t -test for the linear fitting. T -statistic for each coefficient tests the null hypothesis that the corresponding coefficient is zero against the alternative that it is different from zero, given the other predictors in the model.

Fitting Effect	β_{bx}	β_{by}	β_{gx}	β_{gy}	$f(\beta)$
Random 95% Corr. Coeff.	0.7131	0.7172	0.7061	0.7141	0.4363
Tested Term Corr. Coeff.	0.7295	0.7295	0.7295	0.7295	0.7295
Random 95% Rel. Err.	37.15%	36.25%	37.86%	36.32%	55.18%
Tested Term Rel. Err.	36.47%	36.47%	36.47%	36.47%	36.47%
p -value	$<10^{-4}$	$<10^{-4}$	$<10^{-4}$	$<10^{-4}$	$<10^{-4}$
t -statistic	40.925	35.501	40.865	40.076	172.42

Fig.R1. (a) Reconstructed global map of I_d only by planetary beta. (b) Reconstructed global map of I_d with all of the five parameters.

Reviewer #3 (Remarks to the Author):

I am delighted to observe that the author has conscientiously addressed my comments and suggestions in their revised manuscript. Their modifications have considerably enhanced the clarity and coherence of the paper. This work holds substantial significance in advancing our understanding of the structure of mesoscale eddies and the eddy-induced heat transport. With the paper now in good shape, I am glad to recommend its publication in *Nature Communications* after addressing the following minor points.

We thank the reviewer for the positive assessment of our study and the support of our paper for publication in *Nature Communications*.

(1) L50 and L250. The abbreviation of EKE is redundantly defined.

The abbreviation EKE is defined as eddy kinetic energy in Line 50.

(2) L333. Indicate the version of the eddy dataset to facilitate reproducibility of your work by readers.

We thank the reviewer to bring up this point. The Mesoscale Eddy Trajectory Atlas Product (META ver3.2) is used in this paper, and its version information has been added in Line 335 in the revised manuscript.

(3) Figures. I understand that the white area means the region does not hold the geostrophic balance. This should be indicated in the caption of the first figure.

We thank the reviewer to bring up this point, and the corresponding statement has been added to the figure captions.

Reference

1. Chelton, D. B., Schlax, M. G., Samelson, R. M. & de Szoeke, R. A. Global observations of large oceanic eddies, *Geophys. Res. Lett.*, **34**: L15606, doi:10.1029/2007GL030812 (2007).
2. Chelton, D. B., Schlax, M. G., & Samelson, R. M. Global observations of nonlinear mesoscale eddies. *Prog. Oceanogr.* **91(2)**, 167-216 (2011a).
3. Chelton, D. B., Gaube, P., Schlax, M. G., Early, J. J., & Samelson, R. M. The influence of nonlinear mesoscale eddies on near-surface oceanic chlorophyll. *Science* **334(6054)**, 328-332 (2011b).
4. Dong, C., McWilliams, J. C., Liu, Y., & Chen, D. Global heat and salt transports by eddy movement. *Nat. Commun.* **5(1)**, 3294 (2014).
5. Gaube, P., Chelton, D. B., Samelson, R. M., Schlax, M. G. & O'Neill, L. W. Satellite observations of mesoscale eddy-induced Ekman pumping. *J. Phys. Oceanogr.*, **45**, 104-132 (2015).